# An asymmetric sheath controls flagellar supercoiling and motility in the leptospira spirochete

**Kimberley H Gibson[1†], Felipe Trajtenberg[2†], Elsio A Wunder[3,4], Megan R Brady[1], Fabiana San Martin[2], Ariel Mechaly[2‡], Zhiguo Shang[1§], Jun Liu[5], Mathieu Picardeau[6,7], Albert Ko[3,4]\*, Alejandro Buschiazzo[2,7]\*, Charles Vaughn Sindelar[1]\***

[1]Department of Molecular Biophysics and Biochemistry, Yale School of Medicine, New Haven, United States; [2]Laboratory of Molecular and Structural Microbiology, Institut Pasteur de Montevideo, Montevideo, Uruguay; [3]Departament of Epidemiology of Microbial Diseases, Yale School of Public Health, New Haven, United States; [4]Gonçalo Moniz Institute, Oswaldo Cruz Foundation, Brazilian Ministry of Health, Salvador, Brazil; [5]Department of Microbial Pathogenesis, School of Medicine, Yale University, New Haven, United States; [6]Biology of Spirochetes Unit, Institut Pasteur, Paris, France; [7]Integrative Microbiology of Zoonotic Agents, Department of Microbiology, Institut Pasteur, Paris, France

**\*For correspondence:**
albert.ko@yale.edu (AK);
alebus@pasteur.edu.uy (AB);
charles.sindelar@yale.edu (CVS)

[†]These authors contributed equally to this work

**Present address:**
[‡]Crystallography Platform, Department of Structural Biology and Chemistry, Institut Pasteur, Paris, France; [§]Bioinformatics Department, UT Southwestern Medical Center, Dallas, Dallas, United States

**Competing interests:** The authors declare that no competing interests exist.

**Abstract** Spirochete bacteria, including important pathogens, exhibit a distinctive means of swimming via undulations of the entire cell. Motility is powered by the rotation of supercoiled 'endoflagella' that wrap around the cell body, confined within the periplasmic space. To investigate the structural basis of flagellar supercoiling, which is critical for motility, we determined the structure of native flagellar filaments from the spirochete *Leptospira* by integrating high-resolution cryo-electron tomography and X-ray crystallography. We show that these filaments are coated by a highly asymmetric, multi-component sheath layer, contrasting with flagellin-only homopolymers previously observed in exoflagellated bacteria. Distinct sheath proteins localize to the filament inner and outer curvatures to define the supercoiling geometry, explaining a key functional attribute of this spirochete flagellum.

## Introduction

The spirochetes are gram-negative bacteria with helically coiled cells, and encompass a diverse phylum that includes the agents of leptospirosis (*Leptospira* spp.), syphilis (*Treponema pallidum*) and Lyme disease (*Borrelia burgdorferi*). Spirochetes have powerful swimming capabilities that enable them to rapidly disseminate through connective tissue, blood, and organs (*Wunder et al., 2016b*). This swimming capability is enabled by a unique configuration of the flagellum in these organisms (*Charon et al., 2012*; *Li et al., 2000a*; *Wolgemuth et al., 2006*; *Figure 1A*), which allows them to 'drill' through highly viscous media (*Li et al., 2000b*; *Wolgemuth et al., 2006*). By housing their flagella in the periplasm, spirochetes protect these filaments from environmental insults, such as immune surveillance (*Vernel-Pauillac and Werts, 2018*) (a similar function may be achieved by the membranous sheath that coats the exoflagellum in *Vibrio cholerae Yoon and Mekalanos, 2008*). Spirochetal flagella exhibit supercoiled architectures (*Charon et al., 1992*; *Dombrowski et al., 2009*), which have been linked to productive motility (*Wolgemuth, 2015*), and amongst pathogenic species, to virulence (*Ko et al., 2009*; *McBride et al., 2005*; *Sultan et al., 2013*; *Wunder et al., 2016a*).

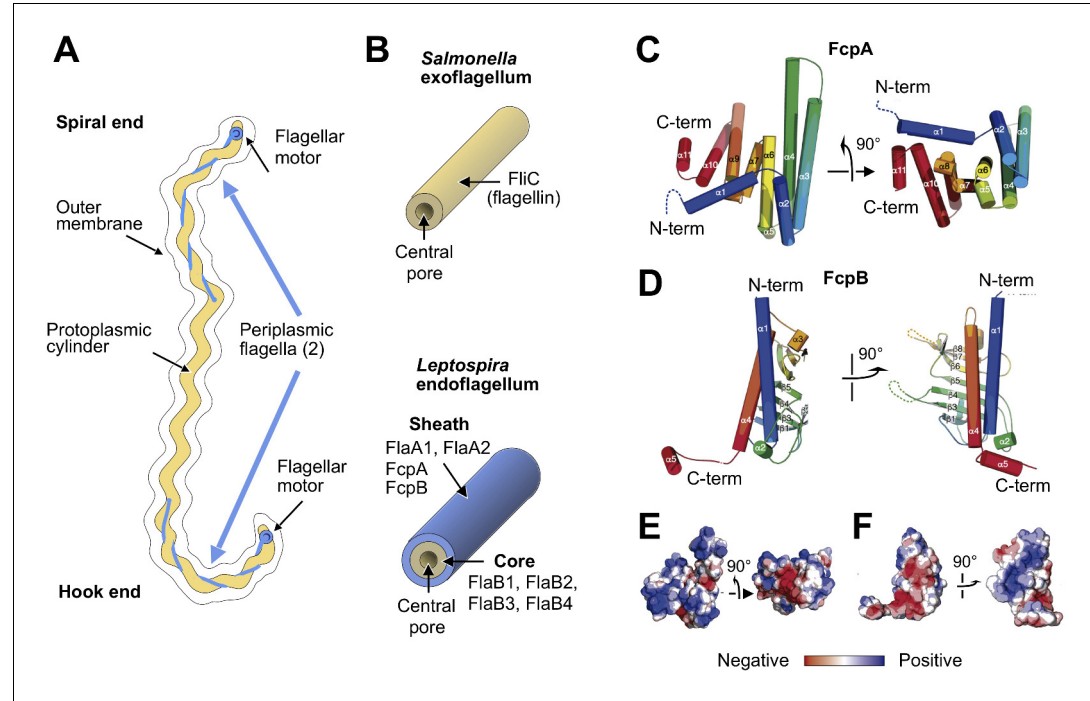

**Figure 1.** X-ray crystal structures of the sheath proteins, FcpA from *L. biflexa* and FcpB from *L. interrogans*. (**A**). Schematic of a *Leptospira* cell. Each cell has two flagella sandwiched between the inner and outer membranes; a single flagellum extends from a motor at either end. (**B**) Comparison of the predicted endoflagellar filament composition and morphology in *Leptospira* with the *Salmonella* exoflagellum. (**C**) Structure of *L. biflexa* FcpA in two orthogonal views, depicted as a cartoon colored with a ramp from blue (N-terminus) to red (C-terminus). The dotted line stands for the flexible 54 amino acids at the N-terminus, not visible in electron density. (**D**) Structure of *L. interrogans* FcpB in two orthogonal views, similar color code as panel A. (**E**) Solvent accessible surface of FcpA colored according to an electrostatic potential ramp from negative (red) through neutral (white) to positive (blue) potentials. (**F**) Solvent accessible surface of FcpB, similar color code as panel E. The perspective was chosen to show the interacting surface of FcpB (positively charged region, panel F left-hand side), in an open-book view (approximately 180° rotated according to a vertical axis), with FcpA (negative cleft, panel E right-hand side). This interaction was later uncovered by studying the whole filament assembly (see *Figure 4E*).

The online version of this article includes the following figure supplement(s) for figure 1:

**Figure supplement 1.** Preferred orientation of purified coiled *L.biflexa* flagellar filaments in the specimen ice layer.

**Figure supplement 2.** Square matrix plot illustrating pairwise sequence identities of flagellar filament proteins from *Leptospira*.

---

*Leptospira* spp. includes pathogenic species that cause human and animal disease, as well as free-living saprophytic species (*Picardeau, 2017*). Unique among known bacteria, including spirochetes, *Leptospira* possess flagellar filaments that spontaneously curl into tightly supercoiled shapes when isolated, following nearly coplanar paths (*Bromley and Charon, 1979*; *Figure 1—figure supplement 1*). The flagellar filament superposes a curved path on the underlying helical body, resulting in overall 'spiral' or 'hook' cell end shapes depending on the motor rotation direction (*Charon and Goldstein, 2002*; *Figure 1A*). Similar to other spirochetes, *Leptospira* cells travel rapidly and unidirectionally when flagellar motors at opposing ends of the organism rotate in opposite directions (i.e. counterclockwise *vs.* clockwise when viewing the motor from the cell exterior), but stall when the motors are both rotating in the same direction (*Goldstein and Charon, 1988*; *Wolgemuth, 2015*). *Leptospira* can also adopt a distinct 'crawling' mode while bound to surfaces (*Tahara et al., 2018*).

As with all spirochetes, *Leptospira* have a complex flagellar composition compared to exoflagellated bacteria (*Charon and Goldstein, 2002*; *Figure 1B*; *Table 1*; *Figure 1—figure supplement 2*). Flagellar filaments from non-spirochete bacteria generally have a single protein component (*Erhardt et al., 2010*) (flagellin). In contrast, spirochetal flagellar filaments are comprised

**Table 1.** Protein components of the flagellar filament from *Leptospira*.

| Protein | Mol. weight | Putative localization in spirochete filaments | Gene names *Leptospira biflexa* (Uniprot ID) | Gene names *Leptospira interrogans* (Uniprot ID) | Protein copies per cell in *L. interrogans* (*Malmström et al., 2009*) | Sequence homologies |
|---|---|---|---|---|---|---|
| FlaB1 | ~31 kDa | Core | LEPBIa2133 (B0SSZ5) | LIC11890 (Q8F4M3) | ~14,000 | • The four FlaB isoforms are homologous to domains D0+D1 of *S. enterica* flagellin FliC.<br>• FlaB1, FlaB2 and FlaB4 are ~ 65–70% identical to each other in both *Leptospira* species.<br>• *L. biflexa* FlaB1 is ~ 87% identical to *L. interrogans* FlaB1. |
| FlaB2 | ~31 kDa | Core | LEPBIa2132 (B0SSZ4) | LIC11889 (Q72R59) | ~2000 | • *L. biflexa* FlaB2 is ~ 78% identical to *L. interrogans* FlaB2. |
| FlaB3 | ~31 kDa | Core | LEPBIa1872 (B0SS86) | LIC11532 (Q72S54) | ~300 | • FlaB3 is ~ 50–55% identical to the other three isoforms in both *Leptospira* species.<br>• *L. biflexa* FlaB3 is ~ 62% identical to *L. interrogans* FlaB3. |
| FlaB4 | ~31 kDa | Core | LEPBIa1589 (B0SQZ5) | LIC11531 (Q72S55) | ~3500 | • *L. biflexa* FlaB4 is ~ 92% identical to *L. interrogans* FlaB4. |
| FlaA1 | ~36 kDa | Sheath | LEPBIa2335 (B0SKT4) | LIC10788 (Q72U74) | ~4500 | • The two FlaA isoforms are not homologous to FlaB or other bacterial flagellins.<br>• FlaA1 and FlaA2 are ~ 25–28% identical in both *Leptospira* species.<br>• *L. biflexa* FlaA1 is ~ 56% identical to *L. interrogans* FlaA1. |
| FlaA2 | ~27 kDa | Sheath | LEPBIa2336 (B0SKT5) | LIC10787 (Q72U75) | ~3500 | • *L. biflexa* FlaA2 is ~ 70% identical to *L. interrogans* FlaA2. |
| FcpA | ~36 kDa | Sheath | LEPBIa0267 (B0STJ8) | LIC13166 (Q72MM7) | ~8000 | • FcpA is unique to the *Leptospira* genus.<br>• FcpA and FcpB are not homologous.<br>• *L. biflexa* FcpA is ~ 77% identical to *L. interrogans* FcpA. |
| FcpB | ~32 kDa | Sheath | LEPBIa1597 (B0SR03) | LIC11848 (Q72RA0) | ~4000 | • FcpB is unique to the *Leptospira* genus.<br>• *L. biflexa* FcpB is ~ 53% identical to *L. interrogans* FcpB. |

of a flagellin homolog (FlaB) and FlaA (*Brahamsha and Greenberg, 1988*), a second and completely unrelated component. FlaA and FlaB proteins can occur in multiple isoforms within a single organism (*Wolgemuth et al., 2006*; *Table 1*). Deletion of FlaA or FlaB in *Leptospira* leads to distinct phenotypes: deletion of FlaA affects flagellar curvature and diminishes motility and pathogenicity (*Lambert et al., 2012*), while deletion of FlaB eliminates the filament entirely (*Picardeau et al., 2001*). Similar observations have been made in other spirochetes (*Li et al., 2000a*; *Motaleb et al., 2000*), indicating that structural and functional adaptations in *Leptospira* flagella may be relevant to spirochetes as a whole. However, *Leptospira* flagella have unique features not found in other spirochetes. Recently, two novel *Leptospira*-specific components of the flagellum, FcpA and FcpB (*Table 1*), were identified whose deletion abolishes the ability of flagella to assume their characteristic supercoiled form and dramatically reduces motility and virulence (*Wunder et al., 2016a*; *Wunder et al., 2018*).

To date, the structure and composition of any spirochetal endoflagellar filament has remained elusive. Here, we present structures of flagellar filaments from the saprophytic spirochete *Leptospira biflexa*, solved by a combination of cryo-tomography, sub-tomogram averaging, X-ray crystallography and molecular docking. Our structures reveal that a conserved FlaB core assembly is enclosed by an asymmetric assembly of sheath proteins with novel folds and functions. Analysis of mutant filaments, which have lost one or more of these sheath elements, indicates that the sheath enforces different core lattice geometries on different sides of the filament, thus promoting a supercoiled flagellar shape. This asymmetric supercoiling mechanism may be relevant to other spirochetes, contributing to their distinctive swimming modes.

# Results

## FcpA and FcpB X-ray structures

Using X-ray crystallography, we established that the flagellar sheath components FcpA and FcpB adopt unique folds, unrelated to globular domains found in available flagellin structures (*Figure 1C–F*; *Galkin et al., 2008*; *Samatey et al., 2001*; *Wang et al., 2017*). FcpA from *L. biflexa* was crystallized in three different space groups (*San Martin et al., 2017*; *Table 2*), revealing a 3D fold previously unseen within the PDB (release July 2019) as reported by several structural alignment algorithms (*Holm and Laakso, 2016*; *Madej et al., 2014*). FcpB from the pathogenic species, *L.*

**Table 2.** X-ray diffraction data processing and model refinement statistics.

| | FcpA_1 | FcpA_2 | FcpA_3 | FcpB |
|---|---|---|---|---|
| Wavelength | 0.97910 | 0.97858 | 0.97858 | 1.54179 |
| Resolution range | 67.36–1.90 (1.94–1.90)* | 48.23–2.95 (3.07–2.95) | 45.12–2.5 (2.6–2.5) | 37.15–2.58 (2.72–2.58) |
| Space group | P 622 | P $2_1$ | C 2 | P $2_12_12_1$ |
| Unit cell (abc Å, αβγ °) | a = b = 132.4 c = 67.4 α=β=90 γ = 120 | a = 85.5 b = 96.5 c = 121.1 α = 90 β = 105.2 γ = 90 | a = 82.3 b = 99.6 c = 106.7 α = 90 β = 91.9 γ = 90 | a = 60.7 b = 65.6 c = 134.4 α = 90 β = 90 γ = 90 |
| Total reflections | 180402 (11851) | 137503 (15749) | 99658 (10749) | 62470 (8683) |
| Unique reflections | 27914 (1742) | 39975 (4485) | 29300 (3278) | 17479 (2442) |
| Multiplicity | 6.5 (6.8) | 3.4 (3.5) | 3.4 (3.3) | 3.6 (3.6) |
| Completeness | 99.7 (99.3) | 99.5 (99.7) | 98.4 (97.9) | 99.2 (97.1) |
| Mean I/sigma(I) | 23.9 (1.5) | 17.2 (2.0) | 18.2 (2.3) | 8.1 (3.2) |
| Wilson B factor | 31.7 | 102.8 | 81.3 | 42.8 |
| R-merge | 0.152 (2.128) | 0.050 (0.651) | 0.040 (0.484) | 0.128 (0.395) |
| R-meas | 0.166 (2.309) | 0.059 (0.768) | 0.047 (0.577) | 0.150 (0.464) |
| $CC_{1/2}$ | 0.991 (0.218) | 0.999 (0.842) | 0.999 (0.805) | 0.993 (0.818) |
| Reflections used in refinement | 27881 (1605) | 39955 (4341) | 29292 (3123) | 17436 (2411) |
| Reflections used for R-free | 1494 (100) | 2005 (208) | 1489 (155) | 1029 (168) |
| R-work | 0.194 (0.259) | 0.197 (0.3445) | 0.196 (0.2586) | 0.204 (0.2739) |
| R-free | 0.219 (0.309) | 0.222 (0.3888) | 0.221 (0.3088) | 0.250 (0.3328) |
| Number of non-hydrogen atoms | 2307 | 7971 | 4089 | 3472 |
| macromolecules | 1979 | 7924 | 3866 | 3410 |
| ligands | 90 | 36 | 84 | 31 |
| solvent | 238 | 11 | 139 | 31 |
| RMS bonds (Å) | 0.010 | 0.010 | 0.010 | 0.010 |
| RMS angles (°) | 0.86 | 1.00 | 1.03 | 1.11 |
| Ramachandran favored [‡] (%) | 98.72 | 98.39 | 98.23 | 95.33 |
| Ramachandran outliers [‡] (%) | 0.00 | 0.00 | 0.22 | 0.00 |
| PDB ID | 6NQW | 6NQX | 6NQY | 6NQZ |
| Raw diffraction data [¶] (doi) | 10.15785/ SBGRID/693 | 10.15785/ SBGRID/691 | 10.15785/ SBGRID/692 | 10.15785/SBGRID/694 (data used to solve the structure by SAD) 10.15785/SBGRID/695 (data used for final structure refinement) |

*Values in parentheses are for highest-resolution shell.

[‡]Calculated by MolProbity [Williams, C. J. et al. MolProbity: More and better reference data for improved all-atom structure validation. Protein Sci 27, 293–315, doi:10.1002/pro.3330 (2018)].

[¶]Deposited in the SBGrid Data Bank public database [Morin, A. et al. Collaboration gets the most out of software. Elife 2, e01456, doi:10.7554/eLife.01456 (2013)].

*interrogans* (ortholog of the *L. biflexa* protein; *Table 1*; *Figure 1—figure supplement 2*), was crystallized in the orthorhombic space group P2$_1$2$_1$2$_1$ (*Table 2*). The resulting fold exhibits distant structural homology to proteins of the Mam33 family, including human mitochondrial protein p32 (*Jiang et al., 1999*) that exhibit affinity for a number of protein partners. Homology to FcpB was also detected with VC1805 (*Sheikh et al., 2008*), a protein from *Vibrio cholerae* that can bind complement protein C1q.

FcpA and FcpB are highly distinct in overall size and shape. FcpA assumes an all-helical fold consisting of a twisted stack of eleven α-helices, arranged in a 'V'-shaped architecture with arms of unequal length (*Figure 1C*). The long arm is formed by an α-helical hairpin structure (α3–4) emanating from a 4-helix bundle (comprising helices α2-3-4-6), while the shorter arm is mainly composed of two 3-helix bundles (α6-7-9 and α9-10-11) bridging to the longer arm via a helical hairpin (α9–10). In contrast to FcpA, FcpB adopts a wide and flat oblong shape, consisting of an 8-stranded anti-parallel β-sheet with two long α-helices (α1 and α4) packed on one side (*Figure 1D*). These major structural differences between FcpA and FcpB are evident even at relatively low resolution (15–20 Å), facilitating their identification in the tomographic reconstructions described below.

## 3D reconstruction of flagellar filaments

The strongly curved supercoiling geometry of intact *Leptospira* flagellar filaments (*Figure 2A*) precluded the use of most structure determination methods, including cryo-EM single-particle 3D reconstruction (*Figure 1—figure supplement 1*, Methods), motivating us to apply a cryo-electron tomography approach. Overlapping series of cubic sub-volumes following curved filament paths were manually selected from 62 reconstructed tomographic volumes from samples of purified, wild-type *L. biflexa* flagellar filaments. A total of ~10,800 sub-volumes were input into a customized subtomogram alignment procedure (*Figure 2—figure supplements 1* and *2*, Materials and methods). The resulting averaged map achieved an overall (average) resolution of ~10 Å (*Figure 2B–E*; *Figure 2—figure supplement 2*) and exhibits a series of concentric protein layers centered around a ~ 2 nm central pore (*Figure 2F*), as well as a distinct array of globular features decorating the filament surface (*Figure 2G*). The morphology of the inner two density layers (*Figure 2G*) closely resembles that of the helically assembled flagellin domains D0 and D1 of filaments from exoflagellated bacteria (*Wang et al., 2017*; *Yonekura et al., 2003*). This structural homology is consistent with earlier predictions that the core of the spirochete filament is composed of FlaB (*Li et al., 2008*; *Nauman et al., 1969*), a protein that lacks the D2 and D3 domains present in flagellins from many exoflagellated bacteria (*Figure 3—figure supplement 1*).

## Asymmetric sheath composition

The tomographic map identified sheath density features that surrounded the two core layers but were not uniformly distributed. Sheath densities were mainly localized on the outer curvature of the filament, or 'convex' side, thus offsetting the filament center of mass away from the central pore when the filament is viewed in cross section (*Figure 2C,F*, *Video 1*). This break in helical subunit arrangement in the *Leptospira* endoflagellum sharply contrasts with symmetric structures which were previously observed for exoflagella from *Salmonella* (*Sasaki et al., 2018*; *Wang et al., 2017*; *Yonekura et al., 2003*), *Campylobacter* (*Galkin et al., 2008*), *Pseudomonas aeruginosa* and *Bacillus subtilis* (*Wang et al., 2017*).

We used two independent docking strategies to fit FcpA and FcpB molecular envelopes into the reconstructed filament map (see Materials and methods) and identified convergent locations for these two proteins that were distributed across the convex side of the sheath region (*Figure 3*, *Video 1*). These sheath sites form an array of globular features that closely follows the canonical 11-protofilament helical lattice identified in flagellar filaments of *Salmonella* and other non-spirochete bacteria (*Figure 3—figure supplement 2*). For FcpA, six adjacent, curved rows of 'V'-shaped density profiles form an interlocking array (*Figure 4*; *Figure 3—figure supplement 3*). The FcpB sites were resolved as four adjacent rows of flattened density lobes protruding slightly outward from the FcpA portion of the sheath (*Figures 3* and *4*). The resulting assembly consists of an inner layer of FcpA molecules in close contact with the FlaB core, and an outer layer of FcpB molecules nestled between rows of FcpA (*Video 2*).

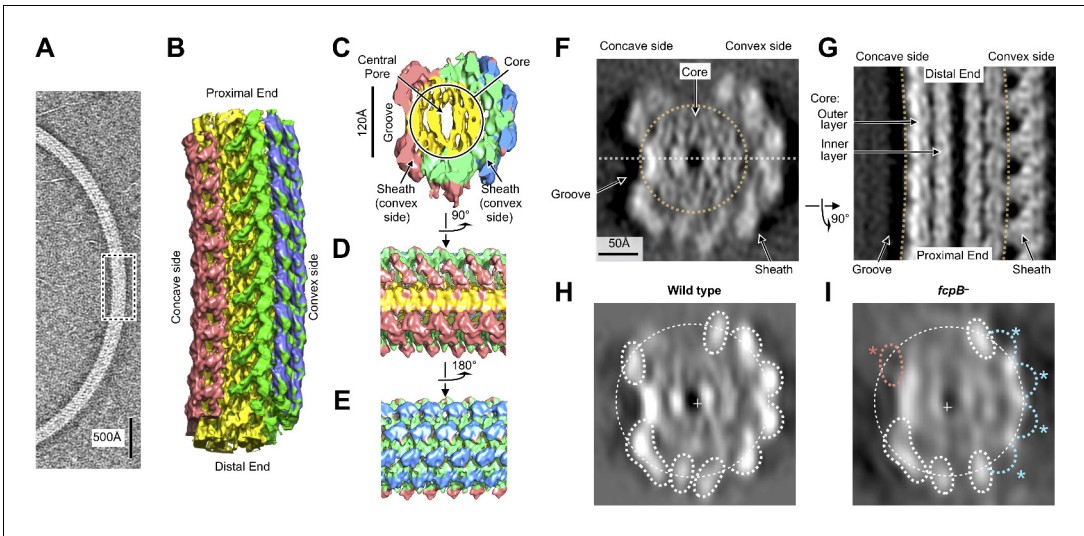

**Figure 2.** Flagellar filament purified from wild-type *L. biflexa* Patoc with asymmetric sheath layer visualized by cryo-tomographic sub-tomogram averaging. (A) Cryo-tomographic slice of the flagellar filament in vitrified ice. Dashed box denotes the approximate dimensions extracted for sub-tomographic averaging. (B) Final averaged volume (isosurface representation) of the flagellar filament denoting segmented regions of sheath (red, green, and blue) and core (yellow). The red sheath regions are located on the inner curvature or 'concave' side of the filament, whereas the blue and green sheath regions are located on the outer curvature, or 'convex' side. (C) Cross-sectional view of the sub-tomographic average; the filament diameter ranges from 210 to 230 Å. (D – E) Rotated views of the sub-tomographic average. (F, G) Wild-type map axial and lateral cross-sections (respectively), highlighting asymmetric features including the 'groove' on the filament inner curvature. The white dashed line in (F) indicates the geometry of the lateral density cross-section in panel (G). (H) Projected wild-type map cross-section, filtered to 18 Å resolution, showing features corresponding to core and sheath elements. (I) Projected *fcpB⁻* map cross-section, highlighting differences with the wild-type projection in H. Four missing densities on the convex side (blue asterisks) correspond to fitted locations of FcpB in the wild-type map; an additional missing density on the concave side (red asterisk) is provisionally assigned as FlaA1 and/or FlaA2 (see text).

The online version of this article includes the following figure supplement(s) for figure 2:

**Figure supplement 1.** Methods flow chart.
**Figure supplement 2.** Progressive improvement of alignment parameters for a representative filament, after various stages of refinement.
**Figure supplement 3.** Resolution estimates for wild-type and *fcpB⁻* subtomogram average reconstructions.
**Figure supplement 4.** Local resolution estimates for the wild-type *L. biflexa* flagellar filament subtomogram average reconstruction.
**Figure supplement 5.** Directional resolution estimates for the wild-type *Leptospira* flagellar filament map, with separate estimates for masked core and sheath subregions.

The asymmetric localization of FcpA and FcpB, specifically on the convex side of the sheath, is distinct from previously characterized exoflagella but is consistent with prior flagellum electron-microscopic imaging studies performed with antibody labels (*Wunder et al., 2016a*; *Wunder et al., 2018*). The localization of FcpB sites were further validated by comparison with a similar reconstruction of filaments from an *fcpB⁻* mutant (*Figure 2I*), for which the projected cross-section of the *fcpB⁻* map was missing four globular features found in the wild-type map (*Figure 2H*) that coincide with our FcpB docking solutions. The signal-to-noise ratio was too low to perform similar sub-tomogram averaging for filaments from an *fcpA⁻* mutant. However, the filaments from imaged *fcpA⁻* mutants were significantly smaller in diameter than filaments from wild-type and *fcpB⁻* mutants (see the last Results section, below), indicating that filaments from the *fcpA⁻* mutant have lost FcpA and FcpB, in agreement with previous findings (*Wunder et al., 2018*).

Additional sheath features are found on the filament concave side, highly divergent from the ones observed on the convex face. These inner-side elements manifested as two rows of elongated density units, separated by a groove that exposes the core (*Figures 2C* and *3A*; *Figure 3—figure supplement 4*). Further modeling of the sheath was precluded as atomic models are not available

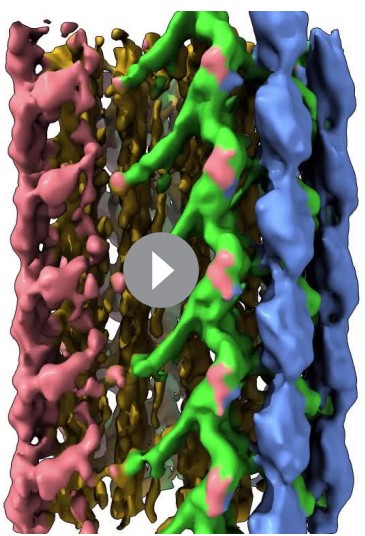

**Video 1.** Overview of the wild-type *Leptospira* flagellar filament map and model, illustrating separately the FcpA and FcpB sheath layers.
https://elifesciences.org/articles/53672#video1

for FlaA1 and FlaA2, the only other known components of the *Leptospira* flagellar filament. FlaA is also presumed to reside in the spirochete sheath (*Li et al., 2008*), and comparing the *fcpB⁻* map with the wild-type map, a missing mass within the concave sheath region indeed suggests candidate positions for FlaA1 and FlaA2 (*Figure 2I*, red asterisk).

The current data do not unambiguously rule out the possibility that FcpA and/or FcpB may occupy one or more sites on the inner curvature. However, a quantitative comparison of core-sheath contact sites for all 11 sites related by pseudo-helical symmetry reveals exactly six protofilaments that share the distinctive 'V'-shaped footprint defined by our FcpA docking studies, while features of corresponding, pseudosymmetry-related sites diverge sharply in the remaining 5 protofilaments (*Figure 3—figure supplement 3B*). Thus, if any of the latter 5 sites are occupied by FcpA and/or FcpB, they must adopt distinctly different binding interfaces with the core. A comprehensive accounting of these features within the concave sheath region awaits further structural and biochemical data.

## Filament core structure

Additional averaging of the map core region revealed homology with flagellar filaments from *Salmonella* and other exoflagellates (*Wang et al., 2017*). Due to a strongly preferred filament orientation in the specimen ice layer (*Figure 1—figure supplement 1*), the potential effect of a 'missing wedge' of Fourier-space data characteristic of the cryo-tomography (*Carazo et al., 2006*) could not be fully eliminated. The resulting map resolution anisotropy (*Figure 2—figure supplement 3*) interfered with structural analyses of the core region, where globular features were neither evident nor expected based on available models of the flagellar core (*Yonekura et al., 2003*). By performing 3D auto-correlation analysis within the core region, however, we detected a signal corresponding to the 11-fold helical symmetry operator present in previously determined flagellar structures (*Figure 3—figure supplement 5*). The 11-fold symmetry operator is mostly congruent with the observed lattice positions of FcpA and FcpB molecules identified in the sheath (*Figure 3—figure supplement 2*), although there are minor deviations due to filament curvature. After applying 11-fold averaging using the symmetry operator, additional features were resolved in the core region, including some α-helices, and revealed a structural homology with the D0/D1 region of the *Salmonella* flagellum (*Figure 3B,C*; *Figure 3—figure supplement 6*; *Video 2*).

Following 11-fold averaging, improved resolution in the core region allowed for straightforward fitting of a core atomic model representing a single 52 Å repeat of FlaB (11 subunits), which we obtained by applying estimated pseudo-helical symmetry parameters to a FlaB monomer homology model (backbone-only atoms) derived from the reported flagellin atomic structure (*Wang et al., 2017*; *Yamashita et al., 1998*; *Yonekura et al., 2003*) (see Materials and methods). The resulting model also shows good agreement with the non-symmetrized map when the superposition is viewed from favorable orientations that minimize missing wedge artefacts (*Figure 3B,C*; *Video 3*).

## Conserved protein-protein interfaces

The resulting filament atomic model exhibits a consistent set of interactions between FcpA and the FlaB core, reflecting the approximate helical symmetry of fitted components (*Figure 4A–C*). FcpA molecules project both arms toward the core region, positioning the long arm within contact radius of two predicted loop regions in FlaB (*Figure 4D*). Such interactions bury the same protein surface

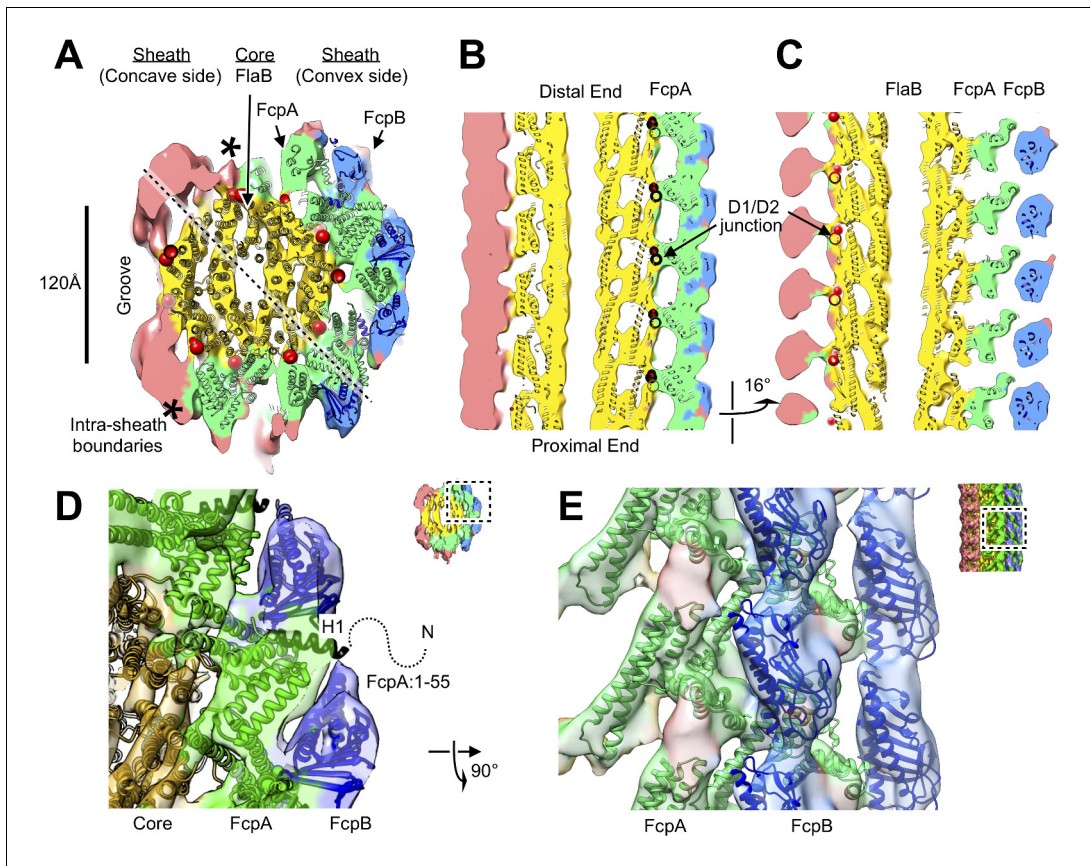

**Figure 3.** Atomic model of the core and sheath regions of the *L. biflexa* flagellar filament obtained by docking X-ray crystal structures into the cryo-EM map. (**A**) Cross-sectional slice of the filament density map isosurface with fitted models of the pseudo-symmetric core assembly (yellow ribbons) and two sheath components, FcpA and FcpB, which localize to the filament outer curvature. Six FcpA protofilaments (green ribbons) directly contact the core and support an outer layer consisting of four FcpB protofilaments (blue ribbons). Asterisks denote boundaries between FcpA and inner curvature density (red). Red markers denote the location of the junction between the modeled D1 α-helical domain of FlaB and the species-specific insertion that substitutes for the D2/D3 outer domains found in *Salmonella* spp. FliC (flagellin) but not in *Leptospira* spp. FlaB. (**B**) Longitudinal slice through the filament center, corresponding to the dashed line in A, showing the central channel surrounded by the core and sheath layers. The major interface region between FcpA and the core coincides with this insertion. (**C**) A 16° rotated view of the map in B showing core-sheath contacts at the site of the FcpB insertions on the opposite side of the filament (concave side); identity of the sheath protein (red) is unassigned. (**D**) Close-up cross-sectional view of the averaged filament map showing X-ray model fits of FcpA and FcpB in the outer curvature sheath region. (**E**) Close-up view of the averaged filament map rotated 90° relative to the view in **D** showing X-ray model fits.

The online version of this article includes the following figure supplement(s) for figure 3:

**Figure supplement 1.** Structure model of *Leptospira* FlaB by homology modeling and sequence alignment.
**Figure supplement 2.** Centroid angular positions of fitted FcpA and FcpB models match an 11-protofilament pseudo-helical lattice.
**Figure supplement 3.** Identification of six similarly arranged FcpA protofilaments in the flagellar filament sheath.
**Figure supplement 4.** Asymmetry of inner and outer sheath density features in wild-type *Leptospira* filaments.
**Figure supplement 5.** Symmetry analysis of the *Leptospira* core indicates an 11-protofilament architecture.
**Figure supplement 6.** Structural homology between the D0/D1 core region from a synthetic map of the *Salmonella* flagellar filament and the core region of our wild-type *Leptospira* flagellar filament subtomogram average volume.

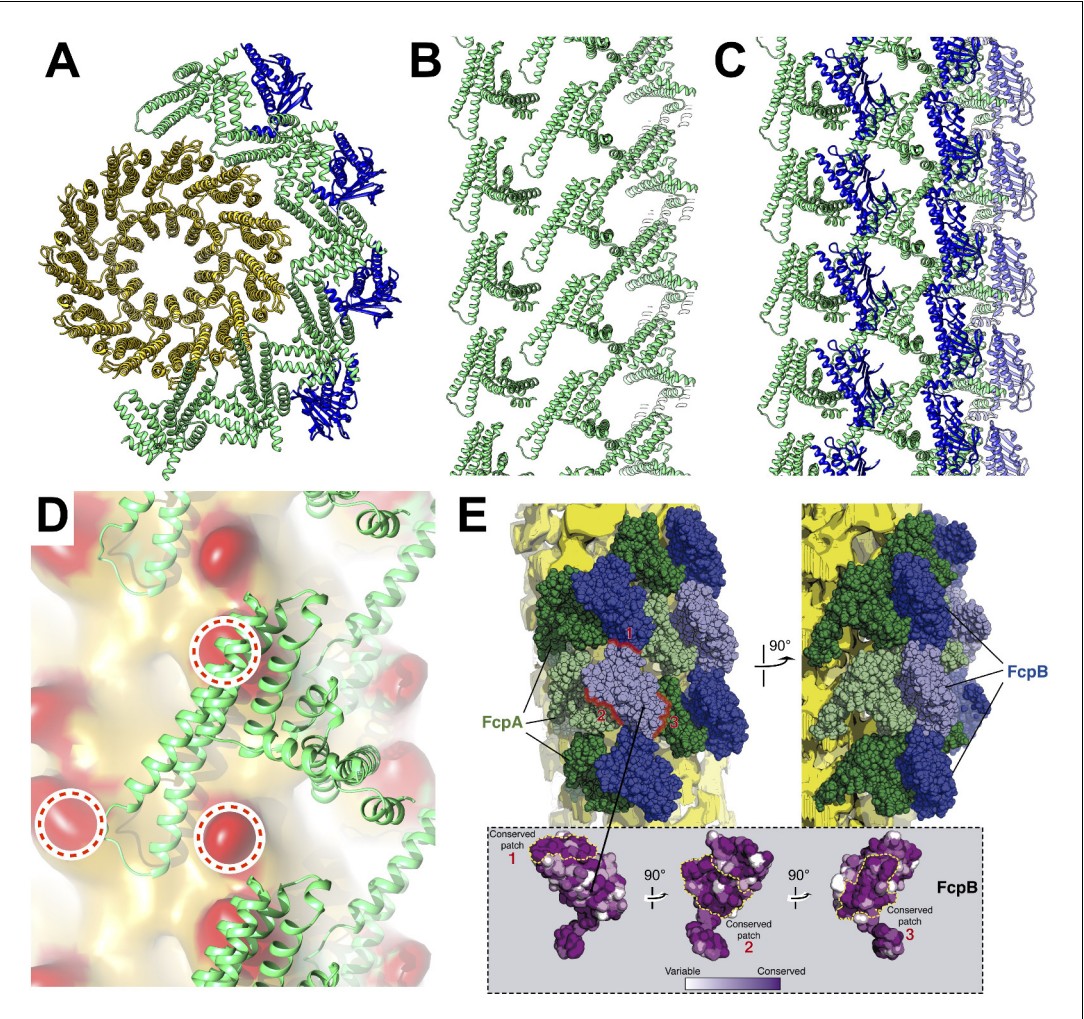

**Figure 4.** Predicted protein-protein interactions from the combined lattice and sheath model. (**A – C**) Ribbon model depiction of the core-sheath atomic model; FlaB is colored gold, FlaA green, FlaB blue. (**A**) is an axial cross-section view, (**B**) is a lateral view of the FcpA lattice only, (**C**) shows the full FcpA/FcpB lattice. (**D**) Predicted interactions between FcpA (green ribbon) and predicted loop regions from FlaB (red, circled). (**E**) Conserved sequence elements on the FcpB surface are oriented towards neighboring FcpA molecules.

of the FcpA α3-α4 hairpin that forms crystal lattice contacts in all three FcpA X-ray structures. The FlaB loop regions engaged in these interactions project outwards from the core surface toward the predicted FcpA arm positions. Thus, FlaB loops in *Leptospira* (both 6–7 residues in length) likely contribute to the functional interface between FcpA and the core.

Modeled FcpB subunits project their central β-sheets radially away from the core and present several elements within contact radius of the FcpA inner sheath layer, including one of the long FcpB α-helices (α4) as well as several loops located at the proximal edge of the β-sheet (*Figure 3D*). We identified two highly conserved surface patches on FcpB which are in close proximity with nearby FcpA molecules (*Figure 4E*), suggesting that these sites may mediate direct interactions between FcpA and FcpB. The corresponding FcpA surface interaction regions predicted by our model are also conserved, and largely match a conserved protein:protein interface found in all FcpA crystal forms. Complementary electrostatic potentials also map onto the FcpA:FcpB interface (*Figure 1E,F*). The modeled FcpB sites are too far away from the core to make direct contact, providing an explanation for why deletion of FcpA leads to concurrent loss of FcpB from the filament (*Wunder et al., 2018*).

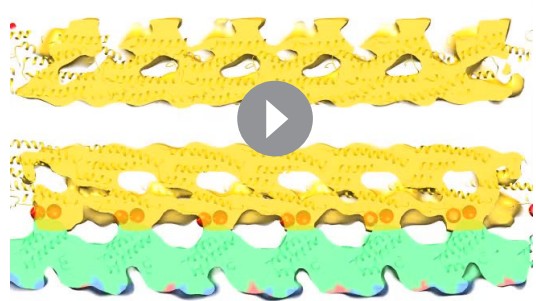

**Video 2.** Cross-section of the filament center, showing superposed wild-type map and atomic model, going through a 360° axial rotation of the filament. The core density is replaced by the 11-fold symmetry-averaged map.

https://elifesciences.org/articles/53672#video2

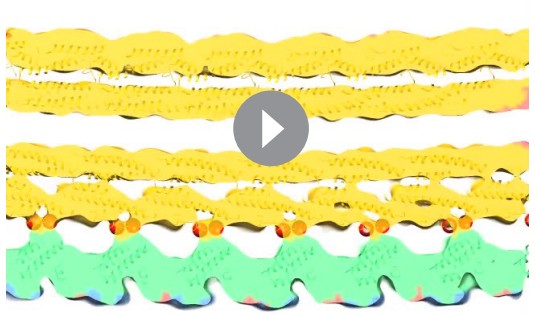

**Video 3.** Similar to *Video 2*, but without the 11-fold symmetry-averaged core. The core atomic model is not fully accounted for by the map density at certain viewing angles, which we attribute to missing wedge artifacts.

https://elifesciences.org/articles/53672#video3

The wild-type filament map provides evidence of contact between FcpA and unassigned elements within the concave sheath region, particularly on one side of the groove (*Figure 3A*; bottom left corner). Since the unassigned material in the concave sheath region is likely composed of FlaA1 and/or FlaA2, FcpA may interact directly with one or both FlaA isoforms. Indeed, pull-down assays of purified *L. biflexa* flagellar filaments recently identified interactions between FcpA and FlaA2 (*Sasaki et al., 2018*).

## The sheath promotes filament curvature

Trajectory analysis of aligned filament subtomograms from wild-type, *fcpA⁻* and *fcpB⁻* samples reveals a progressive straightening in isolated flagellar filaments as sheath components are lost (*Figure 5A–C*). This observation supports and extends previous reports associating these mutations with loss of filament curvature together with pronounced functional effects (*Wunder et al., 2016a*; *Wunder et al., 2018*). Strikingly, straightening observed in the mutant filaments manifests as population shifts between multiple peaks in the observed curvature distributions. Wild-type filament segments almost exclusively fall within a single peak in the measured curvature distribution, centered at the average value (~5 $\mu m^{-1}$; 0.2 $\mu m$ radius of curvature; *Figure 5A*). In contrast, mutant *fcpB⁻* filament segments (which still contain FcpA *Wunder et al., 2018*) show a mixed population of curvatures, with only ~50% of segments falling within the wild-type peak, while curvature for most of the remaining segments is estimated at 3 $\mu m^{-1}$ or less (*Figure 5B*). Curvature is further reduced in *fcpA⁻* filaments, which lack both FcpA and FcpB (*Wunder et al., 2018*): only a vestigial population of segments exhibit the wild-type curvature, while the estimated curvature of most segments is again below 3 $\mu m^{-1}$ (*Figure 5C*).

The *fcpA⁻* filament segments can be divided into two populations, revealing a further correlation between sheath structure and filament curvature. A 'thick' *fcpA⁻* segment population, whose estimated diameter (~170 Å) was consistent with the presence of a partial sheath, shows a propensity to form segments with curvature of ~3 $\mu m^{-1}$ (*Figure 5C*, blue bars). A comparable number of 'thin' *fcpA⁻* segments, with an estimated diameter (~120 Å) indicative of an unsheathed core, shows a population shift towards lower curvature (*Figure 5C*, orange bars). Segment 2D class averages of *fcpA⁻* filaments support these observations, revealing straighter, 'thin' classes indicative of a 'bare' FlaB core, as well as thicker, more curved classes with a visible, partial sheath that may preferentially localize to the inner curvature (*Figure 5—figure supplement 1*). Thus, structure comparisons of wild-type and mutant filaments establish a trend where (1) loss of FcpB converts a substantial fraction of filament segments from the flat, tightly supercoiled, wild-type form to a substantially less-curved form; (2) loss of FcpA and FcpB converts nearly all filaments to less-curved forms; and (3) loss of additional sheath components further straightens the less-curved forms, on average. We occasionally

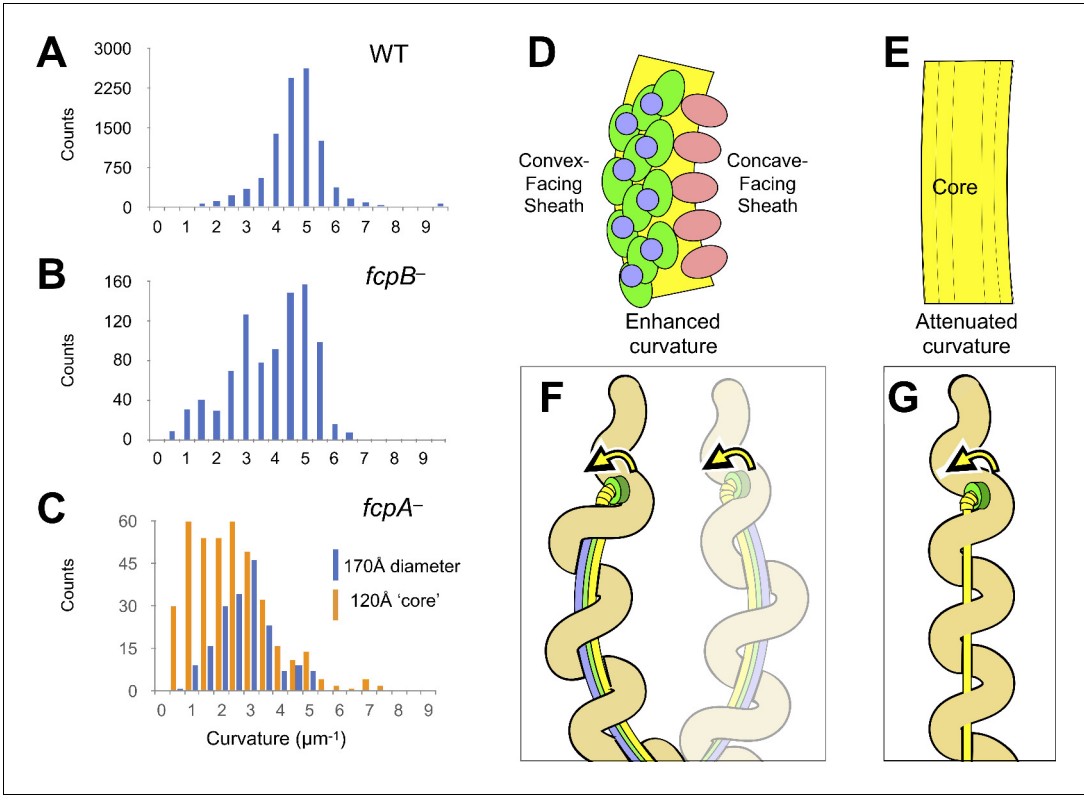

**Figure 5.** The sheath amplifies flagellar curvature to enable motility in the spirochete *Leptospira* spp. (**A**) Histogram of wild-type filament curvatures derived from 3D filament trajectories. A minority of filaments presumed to have shed some or all of the sheath (see *Figure 5—figure supplement 1*), as judged by a smaller measured diameter, were excluded from this analysis. (**B**) Histogram of *fcpB⁻* mutant filament curvatures. As in A, a minority population of smaller-diameter filaments were excluded. (**C**) Histogram of *fcpA⁻* mutant filament curvatures, subdivided into distributions for the larger-diameter population (see *Figure 5—figure supplement 1B*, 3rd panel) and the smaller-diameter population (see *Figure 5—figure supplement 1B*, 4th panel). (**D–E**) Model for sheath-enforced curvature in the *Leptospira* flagellar filament; inherent curvature is amplified due to binding of FcpA and FcpB along the convex side of the core. (**F**) Model depicting how sheath-enforced flagellar curvature would interact with the coiled body in *Leptospira* to generate large-scale curved deformations in the body. (**G**) If flagellar curvature is reduced, the flagellum can pass through the body helix without deforming it, so filament rotation would not directly induce body deformations (except due to rolling and/or sliding friction against the cell cylinder). The online version of this article includes the following figure supplement(s) for figure 5:

**Figure supplement 1.** Shedding of sheath layers observed in wild-type and mutant *L. biflexa* flagellar filaments.
**Figure supplement 2.** Shedding of sheath layers in wild-type *L. biflexa* flagellar filaments coincides with loss of curvature.
**Figure supplement 3.** Supercoiling parameters of wild-type and *fcpB⁻* mutant flagellar filaments estimated from subtomogram average structures.
**Figure supplement 4.** Sinusoidal supercoil forms observed in the *fcpA⁻* sample.

---

observed evidence of the same trend while imaging individual filaments, where an abrupt loss of curvature in a filament coincides with a reduction in filament diameter (indicative of sheath 'shedding'; *Figure 5—figure supplements 1* and *2*).

## Discussion

Here we have identified a novel, asymmetric subunit arrangement in the *Leptospira* flagellum, previously unobserved either in exoflagellated bacteria or in other spirochetes. We have further demonstrated that this asymmetric architecture is functionally linked to flagellar supercoiling, a key attribute associated with spirochete motility. This means of controlling supercoiling contrasts with

factors previously identified in other bacteria, which include changes in buffer composition (salt or pH) or the introduction of torsional strain due to flagellar rotation (*Calladine et al., 2013*). However, earlier investigations of flagellar filament supercoiling have focused on exoflagellates, whose filaments are more simply constructed (homopolymers vs. the heteropolymer considered here), and whose functional roles differ in important ways from spirochete flagellar filaments.

An important innovation in the current work was the application of a subtomogram averaging procedure to solve structures of the supercoiled form present in wild-type *Leptospira* flagellar filaments. This approach contrasts with previous structural studies of flagellar filaments, most of which utilized mutant forms that strongly favor a straight, helically symmetric filament conformation, as was required for high-resolution cryo-EM analysis. One prior study reports the structure of a supercoiled spirochete flagellum (*Liu et al., 2010*); however, in that case the filament was analyzed under the assumption of helical symmetry and was computationally straightened, obscuring any asymmetry. More recently, a supercoiled form of the *Salmonella* flagellar hook was resolved to near-atomic resolution by single-particle cryo-EM (*Kato et al., 2019*; *Shibata et al., 2019*). However, the methods used in these latter works are not well suited for all supercoil morphologies, particularly large-diameter supercoils (as in the current work) that are unfavorably oriented within the specimen ice layer.

The flattened supercoil shape observed in our wild-type *Leptospira* sample, while consistent with prior observations of these filaments (*Wolgemuth et al., 2006*), diverges from forms seen in other bacteria, including exoflagellates (*Calladine, 1975*; *Calladine, 1976*; *Calladine et al., 2013*; *Fujii et al., 2008*; *Kamiya et al., 1980*; *Wolgemuth et al., 2006*). Functional exoflagella, including from *Salmonella*, interconvert between up to 12 distinct supercoil polymorphs, but preferentially adopt an extended, left-handed supercoil form at rest or during smooth swimming ('Normal'; superhelical pitch and diameter of 2.66 and 0.6 μm, respectively). The left-handed, low-pitch geometry we identify in wild-type *Leptospira* filaments (*Figure 5—figure supplement 3*) more closely resembles an alternative exoflagella supercoil type that forms under low pH conditions ('Coiled'; superhelical pitch and diameter of ~0 and 1.0 μm, respectively). However, the supercoil diameter in the *Leptospira* filament (0.45 μm) is less than half that of the 'Coiled' exoflagellar form. Overall, wild-type *Leptospira* filaments are much more curved (curvature is ~5 μm$^{-1}$; *Figure 5A*) than observed or predicted polymorphic forms from exoflagellates, which range in curvature from 0 ~ 3 μm$^{-1}$ (*Calladine, 1975*; *Kamiya and Asakura, 1976*).

Loss of sheath proteins reduces the curvature of *Leptospira* filaments, resulting in supercoiling geometries more consistent with the polymorphic forms observed in exoflagella. Despite their loss of curvature compared with wild-type filaments (*Figure 5A,C*; *Wunder et al., 2016a*), *fcpA⁻* filaments remain markedly more curved than cryo-EM samples of tobacco mosaic virus (TMV) filaments or amyloid fibrils, which lack inherent supercoiling character. For 'straight' TMV and amyloid filaments, an average curvature ~0.7 μm$^{-1}$ was estimated from cryo-EM specimens (attributable to thermal fluctuations, variable fluid flow during sample preparation and other factors) (*Rohou and Grigorieff, 2014*), compared to values of up to ~3 μm$^{-1}$ observed in the *fcpA⁻* sample (*Figure 5C*). Despite loss of most or all of the sheath, the 'thin' *fcpA⁻* population maintains an average curvature value (~2 μm$^{-1}$) that exceeds that of straight TMV or amyloid filaments by more than a factor of two (*Figure 5C*). The estimated curvature for these bare *Leptospira* filament cores, falls within the range observed in exoflagellar polymorphs (0–3 μm$^{-1}$) (*Calladine, 1975*). This characteristic, as well as the occasional observation of 'wavy', sinusoidal *fcpA⁻* filaments of appropriate shape parameters (*Figure 5—figure supplement 4*), supports the idea that the *Leptospira* flagellar filament core may retain polymorphic, supercoiled behavior similar to exoflagellates.

Regardless of specific structural and functional core properties, our data indicate that binding of *Leptospira* sheath proteins (particularly FcpA and FcpB) initiates an allosteric, cooperative transition of the core from a straighter (but potentially polymorphic) form (*Figure 5D,E*), to the flattened, highly curved supercoil geometry suited for swimming (*Figure 5F,G*). A potential mechanism involves recognition by the sheath proteins of distinct core lattice geometries that occur on different sides of a supercoiled filament. For *Leptospira*, these lattice differences are quite substantial: for the given filament geometry (120 Å core diameter and ~0.45 μm superhelix diameter), the axial repeating distance is 3 Å longer on the filament outer curvature than the inner curvature. Therefore, if FcpA preferentially binds to longitudinally expanded protofilaments on the outer curvature of a supercoil, and the core preferentially adopts a shorter longitudinal repeat distance, a helical symmetry mismatch will be generated and a specific supercoiled geometry can be stabilized (*Figure 5F,G*;

*Caspar, 1963*; *Klug, 1967*). In this type of mechanism, binding of an additional sheath component (e.g., FcpB) that is specifically matched to the expanded FcpA lattice would be ideally suited to further reinforce the supercoiling geometry, as is observed here (*Figure 5A–B*).

A noteworthy feature of the flagellar supercoiling mechanism presented in *Figure 5* is that the asymmetric sheath would lock the core into a single, nondegenerate shape. In contrast, a supercoiled, homopolymer flagellar filament can potentially interconvert between 11 equivalent conformations with identical shape parameters but different, rotated patterns of expansion and contraction in the 11 protofilaments. Thus, a supercoiled homopolymer flagellum could propagate a helical wave without rotating (*Klug, 1967*) or equivalently, rotate without propagating a helical wave; an asymmetrically assembled heteropolymer flagellum as we describe for *Leptospira* would be unable to do so.

Unique attributes of *Leptospira* endoflagellar filaments, compared with exoflagella, could be relevant to the different mechanical requirements of these two, divergent propulsion systems. Physics modeling studies indicate that curved filament morphology and stiffness are critical for motility in *Leptospira* and spirochetes in general (*Berg et al., 1978*; *Dombrowski et al., 2009*; *Kan and Wolgemuth, 2007*). Both of these properties can be enforced via the addition of an asymmetric sheath surrounding the filament core. To support the large loads involved in driving whole-body undulations, spirochete endoflagella possess specialized reinforcements, including extra-large motors capable of exerting higher torques than conventional exoflagellar motors (*Beeby et al., 2016*) and unique hooks exhibiting covalent crosslinking between FlgE subunits (*Miller et al., 2016*). Moreover, *Leptospira,* alone among the known spirochetes, has only a single flagellum at each cell end, whereas *Borrelia, Treponema,* and other species exhibit multiple copies per cell end (*Charon et al., 2012*). One function of the sheath elements FcpA and FcpB, to date found only in *Leptospira*, may therefore be to reinforce the endoflagellum to allow for even higher torque loads required in the absence of additional, supporting flagella (*Trachtenberg et al., 1986*; *Trachtenberg et al., 1987*).

Although FcpA and FcpB play conspicuous roles in supercoiling, our images of $fcpA^−$ filaments indicate that the remaining sheath components are themselves capable of asymmetric binding, and may do so preferentially on the concave side of the filament (*Figure 3*; *Figure 5—figure supplement 1*; *Table 3*). These sheath proteins include FlaA1 and FlaA2, although we are not currently able to localize these in the filament map. Orthologs of *Leptospira* FlaA proteins are found in all spirochetes (*Table 1*). Furthermore, loss of FlaA has been shown to modify the diameter (*Li et al., 2000a*) and superhelical pitch (*Lambert et al., 2012*) of spirochetal filaments. These observations raise the possibility that asymmetric arrangement of components in the sheath (and/or the core) may be a general feature that contributes to flagellar supercoiling and motility in spirochetes.

## Materials and methods

### Strains and cell culturing

*Leptospira biflexa* serovar Patoc strain Patoc I (Paris) wild type, $fcpA^-$ and $fcpB$- mutant cells were cultured in Ellinghausen–McCullough–Johnson–Harris (EMJH) liquid medium until they reached logarithmic phase at 30˚C (*Wunder et al., 2016a*; *Wunder et al., 2018*).

**Table 3.** Diameter changes in wild-type and $fcpA^−/fcpB^−$ mutants reflect differences in sheath composition.

| *Leptospira* strain | Inner Curvature[*] | Outer Curvature[†] | Both[‡] | Total # Images |
|---|---|---|---|---|
| WT | 0 | 4 | 7 | 162 |
| $fcpB^−$ | 0 | 4 | 2 | 139 |
| $fcpA^−$ | 16 | 3 | 1 | 83 |

[*] 'Inner curvature' refers to filaments where an abrupt transition in the apparent filament diameter was observed on the concave side of a curved filament.

[†] 'Outer curvature' refers to filaments where an abrupt transition was observed on the convex side.

[‡] 'Both' refers to cases where both 'inner' and 'outer' transitions were observed in the same filament (see *Figure 5—figure supplement 1*).

## Periplasmic flagella purification

Purification of periplasmic flagella was performed as described (*Wunder et al., 2016a*; *Wunder et al., 2018*).

## Recombinant protein crystallization and crystal structure determination

FcpA (Uniprot B0STJ8) from *L. biflexa* serovar Patoc strain Patoc I (Paris) was expressed, purified and crystallized as previously reported (*San Martin et al., 2017*). Three different crystal forms were obtained (*Table 2*), and the structure was solved *ab initio* with the hexagonal data set using Arcimboldo (*Rodríguez et al., 2009*). Resulting electron density maps were automatically traceable (*Lamzin et al., 2012*), and the model was subsequently refined with Buster (Bricogne G. and *Bricogne et al., 2017*) iterated with manual rebuilding and validation with Coot (*Emsley et al., 2010*). The other two monoclinic crystal forms were solved by molecular replacement (*McCoy et al., 2007*) using the hexagonal structure as a search probe, then further refined following similar procedures. The final refined model spans residues 55 to 291, while the first 54 residues toward the N-terminus were disordered.

The gene LIC11848 (Uniprot Q72RA0) coding for FcpB from *L. interrogans* serovar Copenhageni strain Fiocruz L1-130, was cloned in expression plasmid pQE80 (Qiagen), including a TEV-cleavable 6xHis tag. Overexpression was achieved in *E. coli* Rosetta-gami 2 (DE3), induction conditions, chromatographic purification and TEV cleavage procedures were performed as for FcpA (*San Martin et al., 2017*). After cleavage, FcpB started at sequence SQQNSGS (thus excluding the signal peptide), with an N-terminal overhanging tripeptide GSG derived from the plasmid. Single FcpB crystals were grown at 20°C using vapor diffusion technique in hanging drops, mixing 2 µl protein solution (18 mg/mL in 20 mM Tris.HCl pH 8.0, 150 mM NaCl) with 2 µl reservoir solution 0.4M $NH_4I$, 26% (w/v) PEG 3350, 0.05M MES pH 6.5, 21% glycerol. Initial anomalous signal was detectable only to 4.5 Å, improved signal was achieved by quick-soaking crystals in 0.1M 'magic triangle' I3C (*Beck et al., 2008*) for 20 s. Highly redundant data sets were processed using XDS (*Kabsch, 2010*) and scaled with Aimless (*Evans, 2011*). The structure was solved by SAD, exploiting the iodine atoms as anomalous scatterers, the iodine sub-structure was solved with ShelxD (*Schneider and Sheldrick, 2002*) and refined with Sharp (*Bricogne et al., 2003*). Density modification was performed with Pirate and resulting electron density maps allowed for manual chain tracing and initial model refinement with Buster. A second dataset diffracting X-rays to higher resolution was eventually used for final refinement with Buster, iterated with manual rebuilding and validation with Coot.

Structural analyses were done with the CCP4 suite (*Winn et al., 2011*) of programs and with Pymol (*Schrodinger LLC, 2015*).

## Cryo-electron microscopy

*L. biflexa* wild-type flagella. Purified *L. biflexa* wild-type flagella were mixed with 6x concentrated Protein A conjugated with 10 nm colloidal Au (EMS Aurion, Hatfield, PA). Prior to sample application, the 300 mesh Cu Quantifoil grids (Ted Pella, Inc, Redding, CA) with 1.2 µm diameter holes were plasma discharged with $H_2O_2$ for 30 s in a Model 950, Solarus Advanced Plasma System (GATAN). Approximately 3 uL of 2:1 (v/v) flagella: 6x Protein A Gold was applied to each grid. The grids were plunge frozen into liquid ethane in a Mark III Vitrobot (FEI Company, Eindhoven, The Netherlands) following a 2 min incubation time, blot time of 6–7.5 s, and blot offset of −2 mm at 18° C and 100% humidity.

Tilt series were acquired on a 300 kV Polara cryo-TEM (UT Houston, Houston, TX) using SerialEM (*Mastronarde, 2005*) (University of Colorado, Boulder, CO) equipped with a K2 direct electron detector (GATAN, Inc, Pleasanton, CA). Tilt series were acquired with a defocus of between −2.0 and −4.0 µm at +/- 53° with 3° increments in counting mode at a dose rate of ~8 e⁻/pixel/second. At each tilt angle, 12 frames were collected with 0.1 s exposure per frame. The total dosage for each tilt series was ~60 e⁻/Å². The pixel size at a magnification of 15500x was calculated to be 2.604 Å at a binning of 1. A total of 120 tilt series were acquired.

*L. biflexa fcpA⁻* and *fcpB⁻* flagella. Purified *fcpA⁻* and, *fcpB⁻* flagella were prepared similarly to the wild-type flagella and vitrified as described above. Mutant *fcpA⁻* or, separately, *fcpB⁻* flagella were mixed with 6x concentrated 10 nm colloidal Au BSA Gold Tracer beads (EMS Aurion) prior to plunge freezing. The 200 mesh Cu C-flat (EMS) grids with 1.2 µm hole spacing had a 3 nm thick layer of

carbon applied to ensure distribution of gold beads and flagella. The grids were plasma discharged with a 75% argon/25% oxygen mixture for six seconds in a Solarus Advanced Plasma System, Model 950 (GATAN).Tilt series were then acquired on a 200 kV Tecnai F20 cryo-TEM (CCMI, Yale University, New Haven, CT) using SerialEM (*Mastronarde, 2005*) (Boulder, CO) and imaged with a K2 direct electron detector (GATAN) in counting mode. The Tilt series were acquired over +/- 60° in 3° increments with a defocus of −2.5 μm. At a magnification of 14500x at spot size 7, the dose rate was ~8 e⁻/pixel/second. The exposure time per tilt angle was 1.5 s with ~2.25 e⁻/Å² and was dose fractionated into 7 frames, each receiving 0.2 s of exposure time. The final dose was ~90 e⁻/Å² with a pixel size was 2.49 Å. A total of 26 tilt series were acquired.

## Tilt series reconstruction and subtomogram averaging

*L. biflexa* wild-type flagella. The frames in each tilt angle in a tilt series were motion corrected using IMOD alignframes (*Mastronarde and Held, 2017*). The tilt series were then aligned in IMOD, version 4.9.4_RHEL6-64_CUDA8.0 (*Mastronarde and Held, 2017*) using local alignment with the fiduciary markers. After alignment, tilt series underwent CTF-estimation and correction using phase flipping in IMOD, followed by gold subtraction (*Himes and Zhang, 2018*; *Mastronarde and Held, 2017*). Final CTF-corrected and aligned tilt series binned by 2, resulting in a pixel size of 5.208 Å and then reconstructed in Tomo3D (*Agulleiro and Fernandez, 2011*) using weighted back projection. IMOD 3dmod was used to trace a path of particle points along the center of each filament. The addModPts program in PEET (*Cope et al., 2011*) was used to set the repeat spacing between each particle point to 52 Å. Using the IMOD model2point program, the particle model file was converted to a text file format that designated X, Y, and Z coordinates for each filament and assigned an incrementing number for each individual filament for import into RELION version 2.1.b1-gcccuda-2016.10-cc37 (*Scheres, 2012*).

The RELION (*Scheres, 2012*) sub-tomogram averaging module (*Bharat et al., 2015*) was used for initial sub-tomogram alignment. EmClarity (*Himes and Zhang, 2018*) was for subsequent high-resolution refinement and 3D reconstruction, and an in-house method for smoothing and interpolation of sub-volume coordinates (*Huehn et al., 2018*; *Figure 2—figure supplement 2*).

A variation of the RELION pre-processing PYTHON script (*Bharat et al., 2015*) was developed in house to sort particles by filament to ensure splitting of particles for gold standard FSC calculation. This ensured that two adjacent particles in a filament would not be randomly sorted into separate groups, thereby avoiding potential particle coordinate overlap if two or more adjacent particles occupied the same repeat within a filament during averaging and alignment. To ensure minimal polarity switching during particle alignment, larger cubed segments of filaments (500 Å) were extracted and CTF-corrected using CTFFind4. The resulting alignment parameters were imported into RELION version 2.1 beta for unsupervised sub-tomogram averaging using RELION 3D auto-refinement. No mask was used. For 3D auto-refinement, particle dimensions were set to 490 Å to include 490/52 = ~ 9.4 subunits along the filament, and the particle symmetry was designated as C1 due to the inherent asymmetry of the filament. Helical reconstruction options are not implemented for subtomogram averaging in this version of RELION, so the refinement was performed in single-particle mode. An initial angular search spacing of 15° was used. A local search threshold of 1.8° were used for auto-sampling. For local searches, the angular search range in psi and theta (tilt) were 10° and 15°, respectively. The final averaged volume was reported to be 18 Å after FSC calculation during RELION post-processing. Subtomogram averaging in RELION utilized a total of the best 62 tomograms with 10,851 particles.

The flagellum filament structure resulting from subvolume averaging in RELION had a final resolution of 18 Å, but evidently suffered from severe reconstruction artifacts. Efforts to use the RELION subvolume average as a search template for the emClarity (*Himes and Zhang, 2018*) particle picking procedure failed to yield useful particle coordinates. In-house scripts were therefore utilized to export the RELION alignment parameters to emClarity. Following coordinate import, CTF estimation and correction was performed in emClarity. The optimal defocus estimate was set to −3.0 μm and the defocus window was 0.5 μm.

For averaging and alignment steps in emClarity, particle dimensions and box sizes, as well as in-built shape masks were specified in the emClarity parameter file. Following initial estimates for a filament diameter of ~240 Å and a repeat spacing of 52 Å, the particle 'radius' was set to (145 Å, 145 Å, 26 Å) in the parameter file (param0.m). A rectangular alignment mask with x, y, z dimensions of

200, 200, and 160 Å was applied, encompassing 6–8 repeats. Anisotropic SSNR calculation was activated (flgCones = 1). To avoid FSC overestimation due to mixing of closely packed particles together between half datasets, the fscGoldSplitOnTomos option was also activated.

Iterative averaging and alignment steps were run in emClarity for 3–5 cycles with increasingly restrictive search angles and translational shifts. Particle coordinates and Euler angles from several cycles (1, 3 and 5) were extracted from the emClarity results file for analysis of individual filaments. Areas in the tomograms where filaments had lower SNR with respect to the background solvent had poorer tracking of particle coordinates along the filament trajectory, leading to 'breaks' and translational shifts in the x,y,z coordinates or Euler angle misalignments. For each filament, particle coordinates were extracted and graphed using in-house scripts. Particle outliers were deleted, and an in-house script was used to smooth a trajectory through the remaining coordinates, and to fill in gaps in the smoothed trajectory with particles according to a repeat spacing of 52 Å. The particle assessment, excision, and trajectory smoothing procedures were performed several times until the particle coordinates followed a regular, smoothed trajectory. These final coordinates were then re-imported into emClarity for a final 8 cycles of averaging and alignment at a binning of 2 for cycles 0 through 3, and a binning of 1 for cycles 4 through 8. The angular search range (rawAngleSearch=$1^{st}$,$2^{nd}$,$3^{rd}$,$4^{th}$,$5^{th}$) can be set for each alignment cycle with the first two values corresponding to the angular range and increment of the out-of-plane search. The $3^{rd}$ and $4^{th}$ values correspond to the angular range and increment for the in-plane search. The $5^{th}$ value was set to 0 to turn off the helical search, which was not effective for our system. At a binning of 2, the rawAngleSearch range for cycles zero through three, in order, were (0,0,7,1,0), (4,1,0,0,0), (0,0,3,0.75,0), and (2,1,0,0,0).For cycles four through eight, at a binning of 1, the rawAngleSearch ranges were (0,0,5,1,0), (3,1,0,0,0), (0,0,3,0.75,0), (2.25,0.75,0,0,0), and (0,0,1.5,0.5,0). The two half-dataset volumes were combined with a B-factor of 0, and the Gold standard FSC at 0.143 was calculated to be 9.83 A with anisotropic resolutions ranging from 8.89 to 16.17.

*L. biflexa fcpA⁻* and *fcpB⁻* flagella. A similar sub-tomogram averaging procedure as described above was followed for the *fcpA⁻* and *fcpB⁻* flagella tomograms. Specifically, IMOD was used for motion correction (using the program alignframes) and the tilt series were locally aligned through use of the fiduciary markers. CTF correction was also performed in IMOD through phase flipping, followed by subtraction of the fiduciary markers from the tomograms. The final aligned tomograms were binned by 2, resulting in a pixel size of 4.97 Å. Tomograms were then reconstructed in Tomo3D using weighted back projection, and 3dmod (IMOD) was used to select the filaments. The addModPts program in PEET was used to add points corresponding to the repeat spacing of 52 Å, and the IMOD program model2point was used to prepare the files for import into RELION.

The *fcpB⁻* tomograms were imported into RELION using the same scripts as described above, with a box size of 480 Å. A total of 24 tomograms were imported, corresponding to 5917 particles. In the program 3dautorefine, no initial reference model was used, and the symmetry was C1, the particle mask was 470 Å, and the same angular search parameters as for the wild-type structure were used. The resultant *fcpB⁻* structure had a resolution of 26.5 Å, and similar to the wild-type reconstruction, appeared to have large missing wedge artifacts. For *fcpA⁻*, a total of 25 tomograms were used, yielding 4327 particles; particles were imported into RELION and refined using similar parameters. Due to limited signal quality and/or sample heterogeneity, the *fcpA⁻* refinement did not yield a converged structure, with inconsistent particle axial rotations along single filaments. Nevertheless, the resulting *fcpA⁻* subunit x, y, z coordinates yielded continuous trajectories for many of the filaments. Using 2D classification, two distinct *fcpA⁻* particle diameters were identified, with 2958 particles in a smaller-diameter class (~120 Å) and 323 particles in a larger-diameter class (~170 Å). Particle coordinates from these two classes were used for subsequent trajectory/curvature analysis.

For the *fcpB⁻* dataset, the RELION-refined x, y, z coordinates and Euler angles of all 24 tomograms and 5917 particles were imported into emClarity, using the same in-house scripts as described above. CTF correction and estimation was performed, with an initial defocus estimate and window of −5.0 μm and 2.0 μm respectively. Due to reduced signal to noise ratio in the *fcpB⁻* data, the reference alignment procedure failed to unambiguously establish the polarity of many of the filaments, which were therefore discarded. After this analysis, 1163 particles corresponding to 14 filaments from 10 tomograms remained.

Initial emClarity parameters designated the particle radius in x, y, and z as 150, 150, and 20 Å, and applied a rectangular alignment mask of 180,180, 250 Å. As in the wild-type dataset, the

flgCones and fscGoldSplitonTomos options were activated. Five cycles of averaging and alignment were used, in a manner similar to that described above, with each cycle using a binning of 2. The rawAngleSearch range for the cycles, in order, were (0,0,7,1,0), (5,1,0,0,0), (0,0,5,1,0), (4,1,0,0,0), (0,0,3,0.75,0), and (0.75,3,0,0,0). After the fifth cycle, the half data sets were combined, giving an average FSC of 18.41 Å, with a range from 14.58 Å to 31.67 Å at a B-factor of 0.

## Resolution estimation

Initial resolution estimates of final subtomogram average volumes were estimated using emClarity as described (*Himes and Zhang, 2018*; *Figure 2—figure supplement 2*). For additional resolution estimates, emClarity source code was modified to output unmasked copies of both refined half-dataset volumes, and these volumes were input to other software packages: local resolution estimates (*Figure 2—figure supplement 4*) were made using 'Relion version 3.0.6 (*Zivanov et al., 2018*), and direction-dependent resolution estimates (*Figure 2—figure supplement 5*) were made using 3D FSC (*Tan et al., 2017*). For local resolution estimates, maps were sampled at a rate of 40 Å (Relion option '–locres_sampling 40'), with local regions defined by a spherical mask of radius 20 Å softened by a cosine edge function of (40 Å falloff).

Additional masks used in combination with direction-dependent resolution calculations were defined as follows. A curved, quasi-cylindrical mask enclosing the sheathed filament and following the filament path, with a circular cross-section (radius 240 Å), was defined using the 'shape tube' command from UCSF Chimera (*Pettersen et al., 2004*) to generate curved, tubular masking envelopes. To better match the filament cross-section, the intersection of three such masks, slightly offset from each other, was taken to produce a tighter mask with approximately elliptical cross-section. The filament core was defined using the Chimera 'shape tube' command with a radius of 120 Å. These masks were truncated axially to a length of three filament repeats (~106 Å) and then softened by convolution with a spherical cosine edge function (fall-off width of 31 Å).

## Model building, fitting of protein components into tomographic maps and minimization of the *L. biflexa* flagellar filament

Models of *L. biflexa* FlaB1 and FcpB were built by homology modeling with Rosetta[63]. 10,000 models of FlaB1 were thus generated using the structure of the flagellar filament of *B. subtilis* (PDB 5WJT) in a straight symmetric context. In the case of FcpB, we used our *L. interrogans* experimental structure (PDB 6NQZ) as a template, to generate 20,000 models.

Using SITUS (*Kovacs et al., 2018*; *Wriggers, 2012*), the averaged volume of the *L. biflexa* wild-type flagella generated by emClarity was converted to a SITUS readable file using the SITUS map2-map program. Using the colores program, single monomers of FcpA and, separately, FcpB were rigidly fit into the ~10 Å resolution 3D volume with Laplacian filtering using an angular step of 5° where quasi-uniform angular spacing was enforced by the pole sparsing method. The highest-scoring fittings thus obtained gave alignments for two protofilaments of FcpA and one protofilament of FcpB within the outer convex sheath region. To identify additional docking locations within the sheath where resolution anisotropy (due to the missing cone of Fourier data) may have reduced the efficacy of the fitting procedure, we performed additional SITUS searches with the same parameters as before but with modified input maps and models. Specifically, we (i) focused searches on specific sheath regions by segmenting the cryo-EM map into a series of overlapping sub-regions and (ii) increased the signal power of the FcpA and FcpB search models by generating symmetry-related 'triplexes' of the initially discovered docking hits. Sub-region segmentation of the flagellar sheath in our 3D volume was performed using the UCSF-Chimera (*Pettersen et al., 2004*) module Segger. To generate a 'triplex' of FcpA or FcpB atomic coordinates, the highest-scoring initial hit was axially shifted by one 52 Å helical repeat forward and backwards to generate a composite PDB file containing three FcpA (or FcpB) monomers at consecutive sites along a single protofilament. Top-scoring hits from this second round of SITUS searching identified four additional protofilaments of FcpA, for a total of 6 protofilaments, and three additional protofilaments of FcpB for a total four protofilaments.

The relative positioning of FcpA and FcpB in our sheath model was further supported by an independent computational docking strategy, in which we utilized pre-formed FcpA/FcpB dimers predicted by computational methods. In order to construct an FcpA:FcpB complex, we first modeled *L.*

*biflexa* FcpB using our *L. interrogans* experimental structure (PDB 6NQZ) as a template, using standard homology modeling procedures from Rosetta (*Das and Baker, 2008*). Through this procedure, more than 20,000 models of FcpB were generated, including loops β3β4 and β7β8, absent in the experimental structure because of poor electron density in those regions. The best 100 FcpB models, according to the Rosetta score, were used as input for an exhaustive full docking procedure with our FcpA X-ray model. The best 5000 docked configurations, according to the binding energy score, were rigidly fit into a segmented portion of the electron density map of the flagellar filament using SITUS colores and ranked according to the correlation coefficient. The highest-scoring solutions from this search clustered in two distinct molecular arrangements, one of which yielded a fair approximation of a single FcpA/FcpB heterodimer from our full FcpA/FcpB sheath model.

## Curved FlaB core filament building and fitting

To build a complete atomic model of the curved filament, an 11-mer of modeled FlaB subunits (co-assembled according to (straight) helical symmetry) was first fit into the pseudo-helically symmetrized core volume. The 11-mer was then deformed to follow a curved path that matched the measured curvature of the tomographically averaged wild-type filament, using UCSF Chimera (*Pettersen et al., 2004*). This core model, corresponding to a single 52 Å repeat of the filament, was then merged with the best SITUS solutions for FcpA (6 protofilaments) and FcpB (4 protofilaments). The resulting core-sheath atomic model, representing a single 52 Å filament repeat, was then replicated 7 times along the same curved path to generate an assembly representing 8 52 Å repeats. To refine this filament model and eliminate minor clashes, all the protein monomers (protomers) placed in density were subjected to iterative cycles of rigid body, side-chain and backbone minimization using positional and conformational constraints, restrained within the cryoET volume map using a customized protocol in Rosetta (*Das and Baker, 2008*).

## Curvature analysis

Curvature was estimated for each reconstructed filament segment by comparing the 3D coordinates of neighboring filament subunits, and the resulting curvature values summed in histograms. The curvature of the purified flagella was determined in the manner of *Crenshaw et al. (2000)*, utilizing the smoothed three-dimensional coordinates of the filaments. Consider consecutive three-dimensional points $P_1$, $P_2$, $P_3$, ..., $P_N$, with each point spaced 52 Å apart (corresponding to the repeat spacing of the filament). Let $C = P_i - P_{(i-10)}$ and $D = P_{(i+10)} - P_i$ from i = 11 to i=(N-10); a spacing of 10 is used to minimize noise that might result from minor deviations between consecutive points. Then, curvature (κ) at a particular point can be determined by:

$$\kappa_i^* = \left( \frac{C \cdot D}{|C||D|} \right) \left( \frac{2}{|C| + |D|} \right)$$

The final curvature value is then found by averaging the curvature at successive points, in the following manner:

$$\kappa_{<i>}^* = \frac{\kappa_i^* + \kappa_{i+1}^*}{2}$$

The resultant values are multiplied by 1/(pixel size * binning), giving a curvature with units of 1/Å for points 11 to N-10 along the filament. These values were calculated for the 84 filaments of wild-type that were used in the emClarity analysis, the 14 filaments of the *fcpB⁻* sample that were used in the emClarity analysis, and 8 filaments of the *fcpA⁻* sample. Due to the small size of the dataset and sample heterogeneity, a high-resolution structure of the *fcpA⁻* sample could not be determined.

## Estimating supercoil parameters from subtomograms

For each subtomogram average volume (wild-type or *fcpB⁻*), the volume was superposed on a 53 Å axially-shifted copy of itself using the UCSF Chimera command 'Fit in Map'. Parameters for the superhelical rotation axis were then estimated using the Chimera command 'measure rotation'. The superhelical diameter was estimated as twice the distance from the superhelical axis to the filament central channel, while the superhelical pitch was estimated as the reported volume shift along the superhelical axis multiplied by 360° divided by the reported volume rotation.

## Acknowledgements

We gratefully acknowledge Garrett Debs for help with the RELION subtomogram averaging pre-processing step, including his customizations of published python scripts for Relion sub-tomogram averaging. We thank the IT Department from Institut Pasteur for protein modeling computations on the TARS cluster and the staff in the Yale High- Performance Computing facility for their maintenance of these facilities. We thank Prof. Felix Rey for providing insightful discussions about the manuscript.

## Additional information

### Funding

| Funder | Grant reference number | Author |
|---|---|---|
| National Institutes of Health | R01 GM 110530 | Kimberley H Gibson<br>Megan R Brady<br>Zhiguo Shang<br>Charles Vaughn Sindelar |
| National Institutes of Health | U01 AI 088752 | Elsio A Wunder<br>Albert Ko |
| National Institutes of Health | R01 TW009504 | Elsio A Wunder<br>Albert Ko |
| National Institutes of Health | R01 AI052473 | Elsio A Wunder<br>Albert Ko |
| National Institutes of Health | R0 AI121207 | Elsio A Wunder<br>Albert Ko |
| Agencia Nacional de Investigación e Innovación | FCE_3_2016_1_126797 | Felipe Trajtenberg<br>Fabiana San Martin<br>Ariel Mechaly<br>Alejandro Buschiazzo |
| Agencia Nacional de Investigación e Innovación | ALI_1_2014_1_4982 | Felipe Trajtenberg<br>Fabiana San Martin<br>Ariel Mechaly<br>Alejandro Buschiazzo |
| Agencia Nacional de Investigación e Innovación | FCE_3_2016_1_126797 | Mathieu Picardeau |
| Agence Nationale de la Recherche | ANR-18-CE15-0027-1 | Mathieu Picardeau |
| Agence Nationale de la Recherche | ANR-08-MIE-018 | Mathieu Picardeau |
| Pasteur International Joint Research Unit | Integrative Microbiology of Zoonotic Agents (IMiZA) | Mathieu Picardeau |
| Institut Pasteur | | Mathieu Picardeau |

The funders had no role in study design, data collection and interpretation, or the decision to submit the work for publication.

### Author contributions

Kimberley H Gibson, Data curation, Formal analysis, Investigation, Visualization, Writing - original draft, Writing - review and editing; Felipe Trajtenberg, Conceptualization, Data curation, Formal analysis, Validation, Investigation, Visualization, Methodology, Writing - original draft, Project administration, Writing - review and editing; Elsio A Wunder, Conceptualization, Formal analysis, Supervision, Investigation, Methodology, Project administration; Megan R Brady, Software, Formal analysis, Validation, Investigation, Visualization, Methodology, Writing - original draft, Writing - review and editing; Fabiana San Martin, Ariel Mechaly, Zhiguo Shang, Investigation; Jun Liu, Conceptualization, Resources, Formal analysis, Investigation, Methodology; Mathieu Picardeau, Conceptualization; Albert Ko, Conceptualization, Resources, Supervision, Funding acquisition, Writing - original draft,

Project administration, Writing - review and editing; Alejandro Buschiazzo, Conceptualization, Resources, Formal analysis, Supervision, Funding acquisition, Investigation, Visualization, Writing - original draft, Project administration, Writing - review and editing; Charles Vaughn Sindelar, Conceptualization, Resources, Software, Formal analysis, Supervision, Funding acquisition, Validation, Investigation, Visualization, Methodology, Writing - original draft, Writing - review and editing

## Author ORCIDs

Felipe Trajtenberg ⓘ https://orcid.org/0000-0003-0427-5549
Elsio A Wunder ⓘ https://orcid.org/0000-0002-5239-8511
Ariel Mechaly ⓘ http://orcid.org/0000-0002-5305-7495
Albert Ko ⓘ http://orcid.org/0000-0001-9023-2339
Alejandro Buschiazzo ⓘ http://orcid.org/0000-0002-2509-6526
Charles Vaughn Sindelar ⓘ https://orcid.org/0000-0002-6646-7776

## Decision letter and Author response
Decision letter https://doi.org/10.7554/eLife.53672.sa1
Author response https://doi.org/10.7554/eLife.53672.sa2

# Additional files

## Supplementary files
• Transparent reporting form

## Data availability

Atomic models for three crystal forms of FcpA, and for FcpB, have been deposited in the Protein Data Bank under accession numbers 6NQW, 6NQX, 6NQY, and 6NQZ (respectively). Cryo-EM maps have been deposited in the Electron Microscopy Data Bank (EMDB) under accession number EMD-20504. A pseudo-atomic model consisting of main chain FlaB, FcpA and FcpB docked into the wild-type filament reconstruction has been deposited in the Protein Data Bank (PDB) under the accession number 6PWB.

The following datasets were generated:

| Author(s) | Year | Dataset title | Dataset URL | Database and Identifier |
|---|---|---|---|---|
| San Martin F, Trajtenberg F, Larrieux N, Buschiazzo A | 2020 | Flagellar protein FcpA from Leptospira biflexa - hexagonal form | http://www.rcsb.org/structure/6NQW | RCSB Protein Data Bank, 6NQW |
| Mechaly A, Larrieux N, Trajtenberg F, Buschiazzo A | 2020 | Flagellar protein FcpA from Leptospira biflexa / primitive monoclinic form | http://www.rcsb.org/structure/6NQX | RCSB Protein Data Bank, 6NQX |
| Mechaly A, Larrieux N, Trajtenberg F, Buschiazzo A | 2020 | Flagellar protein FcpA from Leptospira biflexa / ab-centered monoclinic form | http://www.rcsb.org/structure/6NQY | RCSB Protein Data Bank, 6NQY |
| Trajtenberg F, Larrieux N, Buschiazzo A | 2020 | Flagellar protein FcpB from Leptospira interrogans | http://www.rcsb.org/structure/6NQZ | RCSB Protein Data Bank, 6NQZ |
| Gibson KH, Sindelar CV, Trajtenberg F, Buschiazzo A, San Martin F, Mechaly A | 2020 | Rigid body fitting of flagellin FlaB, and flagellar coiling proteins, FcpA and FcpB, into a 10 Angstrom structure of the asymmetric flagellar filament purified from Leptospira biflexa Patoc WT cells resolved via subtomogram averaging | http://www.rcsb.org/structure/6PWB | RCSB Protein Data Bank, 6PWB |
| Gibson KH, Sindelar CV, Trajtenberg F, Buschiazzo A, San Martin F, Mechaly A | 2019 | 10 Angstrom structure of the asymmetric flagellar filament purified from Leptospira biflexa Patoc WT cells resolved via subtomogram averaging | https://www.ebi.ac.uk/pdbe/entry/emdb/EMD-20504 | Electron Microscopy Data Bank, 20504 |

The following previously published datasets were used:

| Author(s) | Year | Dataset title | Dataset URL | Database and Identifier |
|---|---|---|---|---|
| Wang F, Burrage AM, Postel S, Clark RE, Orlova A, Sundberg EJ, Kearns DB, Egelman EH | 2017 | Cryo-EM structure of B. subtilis flagellar filaments N226Y | https://www.rcsb.org/structure/5WJT | RCSB Protein Data Bank, 5WJT |
| Maki-Yonekura S, Yonekura K, Namba K | 2010 | L-type straight flagellar filament made of full-length flagellin | https://www.rcsb.org/structure/3A5X | RCSB Protein Data Bank, 3A5X |

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
