## [Decision Letter]

Thank you for submitting your article "An asymmetric sheath controls flagellar supercoiling and motility in the *Leptospira* spirochete" for consideration by *eLife*. Your article has been reviewed by four peer reviewers, including Edward H Egelman as the Reviewing Editor and Reviewer #1, and the evaluation has been overseen by John Kuriyan as the Senior Editor. The following individuals involved in review of your submission have agreed to reveal their identity: Matthias Wolf (Reviewer #2); Chi Aizawa (Reviewer #3).

The reviewers have discussed the reviews with one another and the Reviewing Editor has drafted this decision to help you prepare a revised submission.

Summary:

This manuscript describes a cryo-EM tomography and X-Ray crystallography analysis of the flagella from *Leptospira*, with a focus on the location an arrangement of FcpA and FcpB. The authors show that these proteins localize to a sheath region in the along the outside curve of the flagellum, and that the core region is composed of FlaB, as has been observed in other spirochete flagella. Overall, this is a well-written manuscript that clearly describes the authors' findings. The localization of these proteins and the differences in flagellar conformation that arises in flagella lacking these proteins is informative about the functional role that these proteins play.

Essential revisions:

The main concern raised was that the interpretation is distorted by limited map resolution and ansotropic resolution of the cryo-ET reconstruction. While this does not necessarily reduce the impact of the story, the conclusions are occasionally pushed too far. It would therefore be prudent to scale back some of the claims based on limited level of detail. These are enumerated below.

Introduction: Why have the authors not mentioned the sheathed flagella of *Vibrio* species at all?

Discussion: Since the FcpA and FcpB complex binds to the convex side of the filament, the filament is responsible for determination of overall shape of the flagellum. Please discuss this point.

Do the sheathed flagella show polymorphism under conditions tested in *Salmonella*? If anything is known, please discuss the role of polymorphism.

Subsection "3D reconstruction of flagellar filaments": "lasso-like supercoiling geometry". This term is incorrect. The noun lasso or lariat refers to a noose at the end of a rope – it does not have a supercoiled geometry. Simply use "supercoiled".

Subsection "Asymmetric sheath composition": "the finding of a break in the helical symmetry". Please do not over-interpret this feature – the authors should omit these far-reaching claims. The observed "break" may well be a result of the resolution anisotropy despite averaging of many subtomograms. As the authors point out themselves, the images suffer from a predominance of side views, resulting in insufficient sampling of data within the missing wedge, which is therefore present in all similar oriented subtomograms. This is further illustrated in Figures 2F, G and Figure 3—figure supplement 5A, which clearly shows much lower resolution in the Z (vertical) direction of this cross section. This is compounded by the typical loss of resolution with increasing radius, as often observed in helical reconstructions. Thus, the apparent merging or absence of subunits in the outer sheath domain is not surprising under these circumstances and should not be explicitly claimed as a feature of the flagellum, but rather attributed to the reconstructed map. The shifted features in the mutant reconstruction in Figure 3G, which has even lower overall resolution, are not convincing and may simply be a consequence of sampling different positions on the helical lattice. Furthermore, the rotational average presented in Figure 3—figure supplement 5C shows clearly, that the inner core (FlgE) is likely to share the 11-fold symmetry of the *Salmonella* flagellum, which also argues that individual features of the asymmetric reconstruction cannot be trusted even at the level of subunit resolution.

The authors are aware of the problem, as they write in the figure legend of Figure 1—figure supplement 1C: "*...Leptospira* flagella are largely restricted to the XY plane (theta ~ 0°) due to confinement within the specimen ice layer, yielding a relatively small number of distinct class average images (< 30). This restriction in the range of theta values prevents a meaningful 3D reconstruction from being obtained."

I suggest simply to omit claims related to these map features across the manuscript.

If the authors insist on this claim, they need to supply appropriate statistics to give credence that they can confidently resolve details at the level of one subunit or better at this location in the map. This could be accomplished by providing a proper local resolution map (e.g. by using the program resmap and halfset reconstructions by splitting the number of subtomograms into two sets).

Subsection "Asymmetric sheath composition": it is not clear what the "additional sheath features" are. While Figure 2C and 3A contain obvious differences, these differences may be artifacts of the reconstruction – see above. Simply say that the limited local resolution did not allow positioning of additional subunit models on the concave side and leave it at that.

Subsection "The sheath promotes filament curvature" paragraph two: This paragraph is a nice discussion how the sheath is not the structural reason of supercoiling itself, but that it contributes to stiffness and curvature of the filament. As such, it should better be moved to the Discussion section.

Discussion end of paragraph one: Asymmetry in flagellar filaments has been addressed in recent papers – i.e. Kato et al., 2019; Shibata et al., 2019; Egelman, NSMB 26, 848-849(2019). Please modify the statement and cite these references.

Discussion paragraph two: "unique hooks exhibiting covalent crosslinking between FlgE subunits" – this reference by Lynch et al. does not describe a crosslinked hook that can support large loads but proposes cross-linking as a novel antimicrobial by inhibiting motility with a crosslinker. Remove this part of the sentence.

Discussion paragraph two: "one function of the sheath elements... may be the reinforce the (endo)flagellum for even higher torque loads". This idea, although worthwhile mentioning, is not new – see work by Beeby et al. or the D3 domain of the campylobacter hook (although not an independent protein). Some references may be adequate.

Figure 5F, G. This model is currently not supported by the data. Please omit.

Relatedly, in the penultimate paragraph of the Discussion: "the asymmetric sheath composition... is ideally suited to trigger flagellar supercoiling, by introducing unequal mechanical distortion on opposite sides of the filament": There is a distinction between a role of the sheath on reinforcing curvature or strengthening stiffness, and "triggering flagellar supercoiling". While the former is supported by experiments in this paper, flagellar supercoiling cannot be caused by identical subunits or associated identical sheath proteins, since identical subunits would result in straight helical tubes. Thus, they must be able to change their conformations dependent on their location on the supercoil. How does the unequal mechanical distortion arise spontaneously? The evidence presented is insufficient to make the claim that the sheath proteins alone are responsible for supercoiling.

Continued, in the final paragraph of the Discussion: Therefore, these claims about the asymmetric arrangement of the sheath components should be toned down or removed. The authors also contradict themselves, since they write in the final paragraph of the Results section that filaments maintain supercoiling even in absence of these components. The final paragraph should be rewritten accordingly.

---

## [Author Response]

Essential revisions:The main concern raised was that the interpretation is distorted by limited map resolution and ansotropic resolution of the cryo-ET reconstruction. While this does not necessarily reduce the impact of the story, the conclusions are occasionally pushed too far. It would therefore be prudent to scale back some of the claims based on limited level of detail. These are enumerated below.Introduction: Why have the authors not mentioned the sheathed flagella of Vibrio species at all?

We thank the reviewers for pointing out this omission. We have added a mention of the *Vibrio* sheathed flagellum in the Introduction.

Discussion: Since the FcpA and FcpB complex binds to the convex side of the filament, the filament is responsible for determination of overall shape of the flagellum. Please discuss this point.Do the sheathed flagella show polymorphism under conditions tested in Salmonella? If anything is known, please discuss the role of polymorphism.

We thank the reviewer for this insightful comment, which is closely connected to other points related to asymmetry, which we address below. We have added several paragraphs to the Discussion that address in more detail the role of filament supercoil shape in recruiting (and being stabilized by) the FcpA and FcpB sheath proteins. In particular, the new Discussion now emphasizes evidence in our data that mutants lacking many or all sheath components still form supercoils, albeit with modified geometries. We further discuss evidence for polymorphism in the *Leptospira* filaments, which although not investigated previously, is hinted at by our data.

Subsection "3D reconstruction of flagellar filaments": "lasso-like supercoiling geometry". This term is incorrect. The noun lasso or lariat refers to a noose at the end of a rope – it does not have a supercoiled geometry. Simply use "supercoiled".

We thank the reviewer for noting this incongruity. We have deleted the term "lasso-like".

Subsection "Asymmetric sheath composition": "the finding of a break in the helical symmetry". Please do not over-interpret this feature – the authors should omit these far-reaching claims. The observed "break" may well be a result of the resolution anisotropy despite averaging of many subtomograms. As the authors point out themselves, the images suffer from a predominance of side views, resulting in insufficient sampling of data within the missing wedge, which is therefore present in all similar oriented subtomograms. This is further illustrated in Figures 2F, G and Figure 3—figure supplement 5A, which clearly shows much lower resolution in the Z (vertical) direction of this cross section. This is compounded by the typical loss of resolution with increasing radius, as often observed in helical reconstructions. Thus, the apparent merging or absence of subunits in the outer sheath domain is not surprising under these circumstances and should not be explicitly claimed as a feature of the flagellum, but rather attributed to the reconstructed map. The shifted features in the mutant reconstruction in Figure 3G, which has even lower overall resolution, are not convincing and may simply be a consequence of sampling different positions on the helical lattice. Furthermore, the rotational average presented in Figure 3—figure supplement 5C shows clearly, that the inner core (FlgE) is likely to share the 11-fold symmetry of the Salmonella flagellum, which also argues that individual features of the asymmetric reconstruction cannot be trusted even at the level of subunit resolution.The authors are aware of the problem, as they write in the figure legend of Figure 1—figure supplement 1C: "...Leptospira flagella are largely restricted to the XY plane (theta ~ 0Â°) due to confinement within the specimen ice layer, yielding a relatively small number of distinct class average images (< 30). This restriction in the range of theta values prevents a meaningful 3D reconstruction from being obtained."I suggest simply to omit claims related to these map features across the manuscript.If the authors insist on this claim, they need to supply appropriate statistics to give credence that they can confidently resolve details at the level of one subunit or better at this location in the map. This could be accomplished by providing a proper local resolution map (e.g. by using the program resmap and halfset reconstructions by splitting the number of subtomograms into two sets).

We thank the reviewer for pointing out these weaknesses in our manuscript. We have carefully addressed these issues in detail.

First, we would like to emphasize that our conclusion of asymmetric composition in the sheath is supported by labeling studies, not just by our 3D structure data alone. Antibody labeling studies of both FcpA and FcpB, reported previously and cited in the text, showed specific localization of these sheath components to the filament outer curvature – directly supporting the asymmetric distribution of FcpA and FcpB indicated by our 3D structure analysis.

To further validate our claim of asymmetric organization in the *Leptospira* flagellar sheath, we have performed several additional analyses:

1) We have conducted a careful evaluation of the local resolution in our wild-type subtomogram average (see the new Figure 2—figure supplement 4). Consistent with the reviewer's observation, we found a loss of resolution in the sheath but it is minor (average resolution is reduced to ~11Ã…). Accuracy of these resolution estimates is directly supported by the ability to resolve features at the scale of up to ~10Ã… in our maps, and that many such features are conserved between different protofilaments of the map. To this point, we have modified Figure 2 and added a new supplemental figure (Figure 3—figure supplement 4) to better highlight the quality of map features throughout the filament cross section, including the sheath.

2) As noted by the reviewers, anisotropy in the map resolution is significant in our data sets. To further quantify the problem, we went beyond the originally reported estimates of the resolution anisotropy (Figure 2—figure supplement 3 in the original submission), and used the 3D FSC tool to estimate resolution anisotropy not only for the entire map but also for isolated core and sheath regions (see the new Figure 2—figure supplement 5). The resolution anisotropy estimates support our conclusion that features on the subunit size scale (~30Ã…) are resolvable in the sheath, even for the worst direction corresponding to the missing cone of data. Again, observable features in our map are entirely consistent with the resolution estimates. The new Figure 3—figure supplement 4A includes an example of a "bad" viewing direction (third panel from top), where features "smeared" in the horizontal direction due to the missing cone of Fourier data. Even in this case, substructure in the sheath can be discriminated, and homology is evident with other, less blurred sheath views; this homology especially clear when comparing the right-hand sheath regions of panel 3 and panels 4-6 (all four of these sheath sites are predicted to have identical composition, with one molecule each of FcpA and FcpB).

3) We have redone our occupancy analysis of potential FcpA binding sites (Figure 3—figure supplement 3) to include cross-correlation comparison of the 11 symmetry-related sites. These calculations reveal two categories of sites, precisely corresponding to our localizations of bound FcpA. Using resampled reference regions from the map itself (rather than the PDB), we show a striking fall-off in reference cross-correlation values when comparing the sites we identify as occupied (reference correlations ranged from 0.69 to 0.97) to the "unoccupied" sites (correlation range of -0.01 to 0.39). Together with our new local resolution estimates, this analysis strongly supports our interpretation that FcpA binding is absent (or at the very least, severely perturbed) on the filament inner curvature.

We have also modified the text to emphasize that our essential claim of asymmetry in the sheath composition does not rely on a positive identification of all features in the sheath (which is not currently possible). The presence of a "groove", over 40Ã… wide, where the sheath is completely absent (fully exposing the core 1/11 of the way, or 30Â°, around the filament) can only mean there is at least one protofilament on the inner curvature that lacks some or all of the sheath components that are present on the outer curvature side of the filament (where there is no groove). In our view, presence of the groove (and other asymmetries) in the sheath complements the above-mentioned antibody studies, providing robust evidence for compositional asymmetry in the sheath.

Subsection "Asymmetric sheath composition": it is not clear what the "additional sheath features" are. While Figure 2C and 3A contain obvious differences, these differences may be artifacts of the reconstruction – see above. Simply say that the limited local resolution did not allow positioning of additional subunit models on the concave side and leave it at that.

We thank the reviewer for pointing out this shortcoming of our previous analysis. We have amended the text to explicitly acknowledge that possibility that these "additional sheath features" could represent additional FcpA and/or FcpB subunits. However, we also realize that our previous manuscript did not do a good job of representing the map quality in the sheath region. We have modified Figure 2 and added a new supplemental figure (Figure 3—figure supplement 4) to better highlight the quality of map features throughout the filament cross section, including the sheath.

In particular, we note that subunit-level features (even resolving ultrastructure within a single subunit) are evident within the sheath region on both sides of the filament (inner and outer curvatures). Moreover, as shown in Figure 3—figure supplement 4, panel 3, individual subunit features can even be resolved when the map is projected in the worst directions (where features are smeared out in one direction due to the missing cone of Fourier data). These observations are complemented by our additional resolution analysis above (Figure 2—figure supplements 4 and 5), which demonstrates that the local resolution in the sheath on the inner curvature side is not substantially worse than on the side where we located FcpA and FcpB.

We further note that there is no room in our model of the outer curvature sheath to put any of the missing sheath elements (FlaA1 and FlaA2) known to exist in the *Leptospira* filament. Since these have to go somewhere, it seems clear they must bind to the inner curvature. We have added additional discussion to highlight these points, which in our view argue strongly that most of the inner-curvature sheath material is something other than FcpA and FcpB – most probably FlaA1 and FlaA2.

Subsection "The sheath promotes filament curvature" paragraph two: This paragraph is a nice discussion how the sheath is not the structural reason of supercoiling itself, but that it contributes to stiffness and curvature of the filament. As such, it should better be moved to the Discussion section.

We thank the reviewer for pointing this out. We have moved the paragraph to the Discussion (and integrated it with additional discussion points)

Discussion end of paragraph one: Asymmetry in flagellar filaments has been addressed in recent papers – i.e. Kato et al., 2019; Shibata et al., 2019; Egelman, NSMB 26, 848-849(2019). Please modify the statement and cite these references.

We thank the reviewer for pointing out this omission. We have amended the Discussion to describe and cite these recent results.

Discussion paragraph two: "unique hooks exhibiting covalent crosslinking between FlgE subunits" – this reference by Lynch et al. does not describe a crosslinked hook that can support large loads but proposes cross-linking as a novel antimicrobial by inhibiting motility with a crosslinker. Remove this part of the sentence.

We thank the reviewer for pointing out the mismatch between this statement and the paper we mistakenly cited. We have replaced the citation with correct one that reports the discovery of crosslinking in FlgE hook subunits of the spirochete *Treponema denticola*.

Discussion paragraph two: "one function of the sheath elements.... may be the reinforce the (endo)flagellum for even higher torque loads". This idea, although worthwhile mentioning, is not new – see work by Beeby et al. or the D3 domain of the campylobacter hook (although not an independent protein). Some references may be adequate.

We thank the reviewer for pointing this out. We added additional citations to a pair of papers from Derosier and Macnab.

Figure 5F, G. This model is currently not supported by the data. Please omit.Relatedly, in the penultimate paragraph of the Discussion: "the asymmetric sheath composition... is ideally suited to trigger flagellar supercoiling, by introducing unequal mechanical distortion on opposite sides of the filament": There is a distinction between a role of the sheath on reinforcing curvature or strengthening stiffness, and "triggering flagellar supercoiling". While the former is supported by experiments in this paper, flagellar supercoiling cannot be caused by identical subunits or associated identical sheath proteins, since identical subunits would result in straight helical tubes. Thus, they must be able to change their conformations dependent on their location on the supercoil. How does the unequal mechanical distortion arise spontaneously? The evidence presented is insufficient to make the claim that the sheath proteins alone are responsible for supercoiling.Continued, in the final paragraph of the Discussion: Therefore, these claims about the asymmetric arrangement of the sheath components should be toned down or removed. The authors also contradict themselves, since they write in the final paragraph of the Results section that filaments maintain supercoiling even in absence of these components. The final paragraph should be rewritten accordingly.

We thank the reviewer for pointing out these gaps in our arguments. To address these points, we have added several paragraphs to the Discussion. In our view, the cartoon representation of our results in Figure 5F, G are now strongly supported by our new data and analysis – as noted above, the measured local resolutions (and corresponding, visible density features) in our maps give robust evidence for asymmetry at the subunit level of the sheath.

As for supercoiling, we agree with the reviewer that our data indicate that sheath-less (or at least, mostly sheath-less) fcpA – filaments still tend to supercoil (and thus, are asymmetric). Our revised Discussion removes the implied suggestion that the sheath "introduces" asymmetry to the core. We have added two citations to classic papers (from Klug and Caspar) that give allosteric mechanisms by which supercoiled filaments can be preferentially stabilized, through the introduction of a helical symmetry mismatch between inner and outer layers of a filament. Our model is a variation on this, where the outer layer is provided by a separate protein (sheath components including FcpA, FcpB and/or FlaA1/2) rather than a different part of the identical core subunits (flagellin/FlaB).